# Gaussian Approximation and Concentration of Constant Learning-Rate Stochastic Gradient Descent

**Ziyang Wei**
Department of Statistics
University of Chicago
Chicago, IL 60637
ziyangw@uchicago.edu

**Jiaqi Li**
Department of Statistics
University of Chicago
Chicago, IL 60637
jqli@uchicago.edu

**Zhipeng Lou**
Department of Mathematics
University of California, San Diego
La Jolla, CA 92093
zlou@ucsd.edu

**Wei Biao Wu**
Department of Statistics
University of Chicago
Chicago, IL 60637
wbwu@uchicago.edu

## Abstract

We establish a comprehensive finite-sample and asymptotic theory for stochastic gradient descent (SGD) with constant learning rates. First, we propose a novel linear approximation technique to provide a quenched central limit theorem (CLT) for SGD iterates with refined tail properties, showing that regardless of the chosen initialization, the fluctuations of the algorithm around its target point converge to a multivariate normal distribution. Our conditions are substantially milder than those required in the classical CLTs for SGD, yet offering a stronger convergence result. Furthermore, we derive the first Berry-Esseen bound – the Gaussian approximation error – for the constant learning-rate SGD, which is sharp compared to the decaying learning-rate schemes in the literature. Beyond the moment convergence, we also provide the Nagaev-type inequality for the SGD tail probabilities by adopting the autoregressive approximation techniques, which entails non-asymptotic large-deviation guarantees. These results are verified via numerical simulations, paving the way for theoretically grounded uncertainty quantification, especially with non-asymptotic validity.

## 1 Introduction

In large-scale optimization and streaming-data applications, online learning has played a crucial role, where stochastic approximation serves as a fundamental ingredient to improve computational efficiency and save memory. However, to facilitate trustworthy AI and reliable decision-making based on stochastic approximation, it is important to understand its inherent variability, especially in finite-sample settings.

This paper considers a popular recursive algorithm in online learning – Stochastic Gradient Descent (SGD), also known as the Robbins-Monro algorithm [Robbins and Monro, 1951], which is widely used due to its memory efficiency, computational simplicity, and algorithmic stability. The convergence and distributional theory of SGD and its variants have been extensively studied [Fabian, 1968; Woodroofe, 1972; Pflug, 1986; Polyak and Juditsky, 1992; Kushner and Yin, 1997; Shamir and Zhang, 2013]. Nevertheless, the uncertainty quantification of constant learning-rate SGD remains partially understood. Though central limit theorems (CLT) have been explored in literature [Pflug, 1986; Dieuleveut et al., 2020], refined theoretical properties such as non-asymptotic Gaussian approx-

39th Conference on Neural Information Processing Systems (NeurIPS 2025).

imation rate and sharp concentration inequalities are lacking. To fill in this gap, we provide the first Berry-Esseen bound and the first Nagaev-type tail probability for constant learning-rate SGD.

Specifically, we are interested in the following optimization problem:

$$\theta^* = \arg\min_{\theta \in \mathbb{R}^d} G(\theta) \triangleq \arg\min_{\theta \in \mathbb{R}^d} \mathbb{E}_{X \sim \Pi} g(\theta, X), \tag{1}$$

where $G$ is the objective function from $\mathbb{R}^d$ to $\mathbb{R}$, and $g(\theta, X)$ is the noise-perturbed loss function. With sequentially arriving i.i.d. random data $X_i \sim \Pi$ for some unknown distribution $\Pi$, SGD updates the estimated model parameter as

$$\theta_i = \theta_{i-1} - \gamma \nabla g(\theta_{i-1}, X_i), \ i \geq 1, \tag{2}$$

where $\nabla g$ is the gradient of $g(\theta, X)$ with respect to the first argument $\theta$, and $\gamma > 0$ is the fixed learning rate.

Asymptotic normality of the SGD iterates $\{\theta_i\}_{i \in \mathbb{N}}$ has been investigated in many works. For example, Pflug [1986] proved the first CLT of SGD with constant learning rate, where the SGD iterates were viewed as a time-homogeneous Markov chain. This provides a traditional framework to address the non-stationarity of constant learning-rate SGD. Under certain regularity conditions, such a Markov chain has been shown to converge to a stationary probability measure $\pi_\gamma$ geometrically fast as the iteration $n$ grows to infinity [Dieuleveut et al., 2020; Merad and Gaïffas, 2023]. Building on this asymptotic stationarity, the more recent works focus on the scaling of this stationary distribution in regard to the learning rate $\gamma$, and establish the CLT as $\gamma \to 0$ [Chen et al., 2022; Wei et al., 2025]. Since the constant learning-rate SGD will not converge to $\theta^*$ but oscillates around it with stochastic order of $\sqrt{\gamma}$ [Dieuleveut et al., 2020], we denote by $\theta_\infty \in \mathbb{R}^d$ the random vector following the stationary measure. The diagram below illustrates the key ingredients of classical analysis.

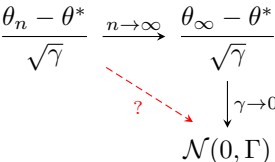

As we can see from the diagram, all the aforementioned works cannot avoid one critical problem: they require the iterations $n \to \infty$ before $\gamma \to 0$. In other words, their asymptotic normality result only holds for the stationary SGD sequence, which, however, is unrealistic in practice, since practitioners usually arbitrarily fix the initialization $\theta_0$ that yields a non-stationary sequence $\theta_i$, while the stationary sequence is unattainable in finite time. In contrast, the actual procedure of stochastic approximation is to first determine a small learning rate, then update the algorithm with a large sample size, which means $n \to \infty$ and $\gamma \to 0$ have to perform concurrently and dependently. Under this circumstance, it remains unclear whether CLT still holds for the SGD iterate itself and, if it does, what constraints between $n$ and $\gamma$ are required. To address the issue, we directly approximate the SGD sequence with iterative linear random functions, skipping the intermediate stationary process. Our methodology exhibits superiority in the sense that, with weaker assumptions, it reveals the simple yet essential relationship between the number of iterations and learning rate sufficient for a quenched version of CLT that holds for any initialization $\theta_0$, and also demonstrates how standardized SGD sequences converge to the normal distribution despite the ordering of limitation on $n$ and $\gamma$. Furthermore, with a slightly stronger condition, linear approximation leads to another powerful finite-sample Gaussian approximation – the Berry-Esseen inequality [Chen and Shao, 2001; Korolev and Shevtsova, 2010; Raič, 2019] – which characterizes the explicit order of distance between SGD and the Gaussian distribution with the number of iterations and learning rate.

Beyond the asymptotic theory, the existing literature on non-asymptotic uncertainty quantification has primarily focused on the linear model [Durmus et al., 2021; Zhu et al., 2022; Agrawalla et al., 2023; Samsonov et al., 2024] or the power-law decaying learning rate [Anastasiou et al., 2019; Shao and Zhang, 2022; Sheshukova et al., 2025], which introduces additional tuning parameters and suffers from surged sensitivity to hyperparameters and slow convergence [Nemirovski et al., 2009; Chee and Toulis, 2018]. Nevertheless, fixed learning rate has recently gained popularity for its simpler tuning requirements and rapid forgetting of the initial value. Moreover, it also enables parallelization of multiple SGD runs to accelerate convergence and one can employ extrapolation techniques for bias

correction [Dieuleveut et al., 2020]. However, establishing non-asymptotic theoretical guarantees for constant learning-rate SGD poses greater challenges due to its persistent oscillations around a stationary region induced by non-diminishing step sizes [Dieuleveut et al., 2020; Cardot et al., 2013, 2017]. To grasp the recursive nature of SGD and comprehend its nonlinear dependence structure, we present a systematic and sophisticated framework based on coupling and dependence measure theory established in Wu [2005]. The high-level idea throughout this paper is to reveal the intrinsic complexity of time series by slight modifications of input data or the iteration mechanism.

## 1.1 Contributions

This paper advances the theoretical understanding of constant learning-rate SGD by introducing novel approximation techniques and deriving sharp finite-sample guarantees. Our main contributions are:

- **A linear-approximation CLT for constant-step SGD.** We develop a new linearization framework that captures the drift and noise dynamics of SGD iterates. Under substantially weaker smoothness and moment conditions than those in Pflug [1986] and Dieuleveut et al. [2020], we prove a central limit theorem showing that, as the step size $\gamma \to 0$, the properly scaled SGD iterate converges in distribution to a Gaussian law.

- **Non-asymptotic $p$-th moment bounds for $p \geq 2$.** Going beyond weak convergence, we derive explicit finite-sample upper bounds on $\mathbb{E}[|\theta_n - \theta^*|^p]$ for any integer $p \geq 2$. The moment convergence rate quantifies how quickly all higher moments of the error decay in $n$, which provides a more refined theoretical guarantee compared to the weak convergence in the probability measure.

- **First Berry–Esseen bound in the Gaussian approximation.** We first provide a quenched version CLT of the last-iterate SGD. Under mild regularity conditions, if $n \to \infty$ and $\gamma \to 0$ with $n\gamma \geq \nu \log n$ for some $\nu \geq 1/(2\gamma)$, then for any initialization $\theta_0 \in \mathcal{R}^d$ the scaled iterate satisfies
$$\frac{\theta_n - \theta^*}{\sqrt{\gamma}} \xrightarrow{D} \mathcal{N}(0, \Gamma),$$
where the matrix $\Gamma$ is later defined in (6). Furthermore, we also obtain the Berry–Esseen type rate (up to logarithmic factors) for the distributional distance between the scaled SGD iterate and its Gaussian limit. We quantify how rapidly the convergence to normality occurs in finite samples and fixed $\gamma$ by providing the rate for
$$\sup_{\mathcal{D} \in \mathcal{V}} |\mathbb{P}(\frac{\theta_n - \theta^*}{\sqrt{\gamma}} \in \mathcal{D}) - \mathbb{P}(\mathcal{N}(0, \Gamma) \in \mathcal{D})|$$
where $\mathcal{V}$ is the collection of all convex sets in $\mathbb{R}^d$. This is the first Gaussian approximation bound for constant-learning-rate SGD in general settings, which allows one to assess the accuracy of statistical inference with finite samples.

- **Nagaev-type large-deviation inequalities.** By approximating the SGD recursion with an autoregressive process, we derive sharp Nagaev-style tail bounds that control the probability of large deviations beyond the CLT regime. These results yield explicit sub-Gaussian and polynomial terms in the convergence rate for $\mathbb{P}(|\theta_n - \theta^*| > \epsilon)$ for any $\epsilon > 0$, which relates to tight sample complexity bounds significant for statistical learning theory and applications [Valiant, 1984].

## 1.2 Other related works

**Stochastic Gradient Descent.** The large-sample behavior of SGD and its extensions dates back to the foundational analyses by Blum [1954]; Sacks [1958], who first investigated its asymptotics and has since been elaborated upon by a succession of studies [Ljung, 1977; Lai, 2003; Wang and Gao, 2010; Gandikota et al., 2022; Zhong et al., 2024; Li et al., 2024]. In particular, Fabian [1968] characterized the limiting law of the final iterate. Robbins and Siegmund [1971] then leveraged martingale arguments to establish almost-sure convergence of the procedure. Subsequent research has quantified the algorithm's convergence rates under a variety of assumptions [Toulis and Airoldi, 2017; Pillaud-Vivien et al., 2018; Muecke et al., 2019; Duchi and Ruan, 2021]. See more recent works by Zhu et al. [2022]; Hu and Fu [2024]; Lauand and Meyn [2024] among the others.

**Convergence with different learning rates.** The learning-rate choice in SGD, either fixed or decaying, critically shapes convergence. Pflug [1986] first studied the stationary behavior under a constant step size via a Markov-chain framework; Dieuleveut et al. [2020]; Huo et al. [2023]; Merad and Gaïffas [2023] later refined this with Wasserstein-distance analyses and high-confidence bounds. For diminishing steps, Rakhlin et al. [2012] showed the linear decaying case attains the optimal rate, and Ge et al. [2019] extended the results to polynomial decays. Burn-in strategies were proposed by Gower et al. [2019] and Nguyen et al. [2019]. Adaptive schemes include Polyak's method for over-parameterized models [Loizou et al., 2021] and a bandwidth-based family studied by Wang and Yuan [2023]. See Jiang and Stich [2024] for a comprehensive overview. However, in practice, constant learning rates are favored due to the simplicity of training, which is also the focus of this study.

**(Non)-asymptotic normality.** In addition to convergence guarantees, performing real-time inference with SGD-style estimators is vital for uncertainty quantification. Classical bootstrap-based M-estimation methods [Fang et al., 2018; Fang, 2019; Zhong et al., 2024] are computationally prohibitive in streaming contexts. Instead, Polyak–Ruppert averaging [Ruppert, 1988; Polyak and Juditsky, 1992] offers statistical efficiency and enables inference: the averaged SGD (ASGD) sequence [Györfi and Walk, 1996; Defossez and Bach, 2015] admits an asymptotic normality result at the optimal rate [Moulines and Bach, 2011; Dieuleveut and Bach, 2016; Dieuleveut et al., 2017; Jain et al., 2018]. By contrast, inference on the final iterate under a constant step size has seen little treatment. We close this gap by proving a quenched central limit theorem for the SGD estimator as the learning rate $\gamma \to 0$, valid from any initialization [Dahlhaus and Rao, 2006; Dahlhaus et al., 2019]. Additionally, blocking-based variance estimators [Chen et al., 2020; Zhu et al., 2023] and recursive kernel approaches [Huang et al., 2014] achieve optimal mean-squared-error rates under dependence, yielding practical, theoretically sound online inference for SGD.

## 1.3 Notation

For a vector $v = (v_1, \ldots, v_d)^\top \in \mathbb{R}^d$ and $q > 0$, we denote $|v|_q = (\sum_{i=1}^d |v_i|^q)^{1/q}$ and $|v| = |v|_2$. For any $s > 0$ and a random vector $X$, we say $X \in \mathcal{L}^s$ if $\|X\|_s = (\mathbb{E}|X|_2^s)^{1/s} < \infty$. For two positive number sequences $(a_n)$ and $(b_n)$, we say $a_n = O(b_n)$ or $a_n \lesssim b_n$ (resp. $a_n \asymp b_n$) if there exists $C > 0$ such that $a_n/b_n \le C$ (resp. $1/C \le a_n/b_n \le C$) for all large $n$, and write $a_n = o(b_n)$ if $a_n/b_n \to 0$ as $n \to \infty$. Let $(X_n)$ and $(Y_n)$ be two sequences of random variables. Write $X_n = o_\mathbb{P}(Y_n)$ if $X_n/Y_n \to 0$ in probability as $n \to \infty$. Let $\langle \cdot, \cdot \rangle$ denote the canonical inner product in the finite dimensional Euclidean space $\mathbb{R}^d$.

## 2 Moment convergence

We first introduce the following assumptions on the objective function $G(\theta)$ and the stochastic gradients $\nabla g(\theta, X)$.

**Assumption 2.1** ($\mu$-strong convexity)**.** *The function $G$ is twice differentiable and $\mu$-strongly convex, i.e. for a $\mu > 0$ and for all $\theta, \theta' \in \mathbb{R}^d$, it holds that*

$$\langle \nabla G(\theta) - \nabla G(\theta'), \theta - \theta' \rangle \ge \mu|\theta - \theta'|^2.$$

Since Assumption 2.1 requires that $G$ is twice differentiable, the Hessian matrix $\nabla^2 G(\theta^*)$ exists. Note that for the $L^p$ convergence in Theorem 2.3, we only need $G$ to be continuously differentiable. The existence of Hessian is only necessary to derive the CLT and Gaussian approximation in Theorem 3.4 and 3.5.

**Assumption 2.2** (Stochastic Lipschitz continuity)**.** *The function $g(\theta, x)$ is continuously differentiable w.r.t. $\theta$ for any $x$. Moreover, for some $p \ge 2$, assume $\|\nabla g(\theta^*, X)\|_p =: M_p < \infty$, $\|\theta_0 - \theta^*\|_p =: \rho_p < \infty$, and the stochastic Lipschitz continuity,*

$$\|\nabla g(\theta_1, X) - \nabla g(\theta_2, X)\|_p \le L_p|\theta_1 - \theta_2|, \quad \text{for all } \theta_1, \theta_2 \in \mathbb{R}^d.$$

Here, the condition $\|\theta_0 - \theta^*\|_p < \infty$ trivially holds in the fixed-initialization setting. Both strong convexity and Lipschitz continuity conditions are commonly adopted in the literature; see for example, Dieuleveut et al. [2020]; Zhu et al. [2022]; Merad and Gaïffas [2023]. Notably, Assumption 2.2 also implies the $L$-smoothness of the gradient of objective function $G$, that is

$$|\nabla G(\theta_1) - \nabla G(\theta_2)| \le L_p|\theta_1 - \theta_2|, \quad \text{for all } \theta_1, \theta_2 \in \mathbb{R}^d.$$

We refer to Li et al. [2024] for a detailed discussion. Now we are ready to present the $L^p$ moment convergence of constant learning-rate SGD.

**Theorem 2.3** ($L^p$ Convergence). *Suppose that Assumptions 2.1 and 2.2 hold. Let $\alpha_0$ be some constant satisfying*

$$0 < \alpha_0 \leq \min\left\{\frac{1}{\gamma}, 2\mu - (6p-5)L_p^2\gamma\right\}. \tag{3}$$

*Then, for any $n \geq 1$ and $\gamma$ satisfying*

$$0 < \gamma < \frac{2\mu}{(6p-5)L_p^2}, \tag{4}$$

*we have,*

$$\|\theta_n - \theta^*\|_p^2 \leq (1-\alpha_0\gamma)^n \rho_p^2 + 3(p-1)M_p^2\alpha_0^{-1}\gamma. \tag{5}$$

**Remark** (Rate of moment convergence). *The right-hand side of (5) demonstrates that SGD with a constant learning rate forgets its initial condition with an exponential pace. Besides, if we determine the learning rate $\gamma$ based on the knowledge of the total sample size or number of iterations $n$, a constraint $n\gamma \to \infty$ should be satisfied to ensure convergence, i.e., the total length of steps needs to be sufficiently large such that SGD can move on. This relationship is quite natural and required by most of the literature on SGD [Polyak and Juditsky, 1992; Kushner and Yin, 1997; Sheshukova et al., 2025]. For example, consider the decaying learning rate schedule $\eta_n \asymp n^{-\alpha}$, it is well-known that $\alpha$ can not exceed 1 for the sake of $\sum_{i=1}^n \eta_i \to \infty$.*

## 3 CLT and Berry-Esseen theorem

In this section, we introduce how to approximate SGD sequences with recursive linear random functions. Consequently, we establish asymptotic normality and finite-sample Gaussian approximation directly on the SGD iterations.

### 3.1 Dependency of SGD iterates

In recursive algorithms, each new updated estimator depends on the last update and the new-coming random sample, which involves an intricate dependency structure that poses a challenge to normality analysis. To address this issue, we leverage the theory of functional dependence measure introduced in Wu [2005] to quantify the dependence structure of the SGD algorithm. Define $\theta_n = \tau_n(X_1, X_2, ..., X_n)$ for some measurable function $\tau_n$ that can vary for different $n$ and $\theta_n^{(t)} = \tau_n(X_1, X_2, ..., X_{t-1}, X_t', X_{t+1}, ..., X_n)$, where $X_t'$ is an i.i.d. copy of $X_t$. The functional dependence measure is defined as

$$\psi(n, t, p) = \|\theta_n - \theta_n^{(t)}\|_p.$$

We denote $\psi(n, t) = \psi(n, t, 2)$ for simplicity. In particular, $\psi(n, t, p)$ quantifies the effect of the random sample $X_t$ on the $n$-th SGD iterate $\theta_n$. By utilizing this tool, we can derive exact Gaussian approximation and tail probability rates. We first provide an essential bound of this dependence measure, which will be heavily used in the proofs.

**Theorem 3.1** (Functional dependence measure). *Suppose the same conditions of Theorem 2.3 hold. The functional dependence measure satisfies*

$$\psi(n, t, p) \leq 2\sqrt{2}\gamma(1-\alpha_0\gamma)^{(n-t)/2}\sqrt{M_p^2 + L_p^2[(1-\alpha_0\gamma)^{t-1}\rho_p^2 + 3(p-1)M_p^2\alpha_0^{-1}\gamma]}.$$

**Remark** (Rate of dependence measure). *Theorem 3.1 encapsulates the temporal dependence of constant learning rate SGD, which decays at an exponential rate as the time lag increases.*

### 3.2 Refined linear approximation and asymptotic normality

We first introduce another assumption only required for the rest of this section.

**Assumption 3.2** (Local Smoothness). *There exists some constants $L \geq 0$ and $\kappa > 0$ such that for all $|\theta - \theta^*| \leq \kappa$,*

$$|\nabla^2 G(\theta) - \nabla^2 G(\theta^*)| \leq L|\theta - \theta^*|.$$

The local smoothness of the Hessian is standard in the literature on statistical inference and normal approximation of online learning algorithms [Anastasiou et al., 2019; Shao and Zhang, 2022; Li et al., 2022; Sheshukova et al., 2025]. Assumption 1 in [Anastasiou et al., 2019] requires a stronger global Lipschitz smoothness. We only need this condition to ensure the validity of Taylor's expansion of the gradient, i.e.,

$$|\nabla G(\theta) - \nabla^2 G(\theta^*)(\theta - \theta^*)| \lesssim |\theta - \theta^*|^2,$$

which is also commonly imposed [Ruppert, 1988; Polyak and Juditsky, 1992] and necessary for Assumption H6 in [Moulines and Bach, 2011].

Define $A := \nabla^2 G(\theta^*)$ as the Hessian matrix of the objective function $G(\theta)$ at $\theta = \theta^*$, and $S = \mathbb{E}[\nabla g(\theta^*, X_n)\nabla g(\theta^*, X_n)^\top]$ as the covariance of the stochastic gradients, also at the true parameter $\theta^*$. Denote the estimation error by $\Delta_n = \theta_n - \theta^*$. We have

$$\Delta_n = \Delta_{n-1} - \gamma A\Delta_{n-1} + \gamma R_n + \gamma D_n - \gamma\nabla g(\theta^*, X_n),$$

where we define the Taylor expansion remainder and the martingale difference noise term respectively by

$$R_n = A\Delta_{n-1} - \nabla G(\theta_{n-1}),$$
$$D_n = \nabla G(\theta_{n-1}) - \nabla g(\theta_{n-1}, X_n) + \nabla g(\theta^*, X_n).$$

Recursively updating the formula, we get

$$\Delta_n = (\mathbf{I}_d - \gamma A)^n \Delta_0 - L_n + I_{1,n} + I_{2,n},$$

where

$$L_n = \gamma \sum_{k=1}^{n} (\mathbf{I}_d - \gamma A)^{n-k} \nabla g(\theta^*, X_k),$$

$I_{1,n} = \gamma \sum_{k=1}^{n} (\mathbf{I}_d - \gamma A)^{n-k} R_k$ and $I_{2,n} = \gamma \sum_{k=1}^{n} (\mathbf{I}_d - \gamma A)^{n-k} D_k$. Due to stochastic Lipschitz continuity and Theorem 5, the terms $I_{1,n}$ and $I_{2,n}$ are infinitesimal of higher order, and the estimation error of SGD can be well approximated by the linear sequence $L_n$. Let $\lambda^* > \lambda_* > 0$ denote the largest and smallest eigenvalue of $A$, and $\lambda = \min\{\lambda_*, \alpha_0\}$. We have the following result,

**Lemma 3.3** (Linear approximation). *Under Assumption 3.2 and same conditions of Theorem 2.3, for*

$$\gamma < \min\{\frac{1}{\lambda^*}, \frac{\mu}{(6p-5)L_p^2}\},$$

*we have*

$$\|\Delta_n - L_n\|_1 \leq \left(\frac{3C_0 M_2^2}{\lambda\alpha_0} + \frac{2\sqrt{3}L_2 M_2}{\sqrt{\lambda\alpha_0}}\right)\gamma + \frac{2L_2\rho_2}{\sqrt{\lambda}}\sqrt{\gamma}(1-\lambda\gamma)^{\frac{n-1}{2}} + C_0\rho_2^2 n\gamma(1-\lambda\gamma)^{n-1},$$

*where $C_0 = \max\{L, 2L_2\kappa^{-1}\}$.*

We will show in the appendix that the scaled limiting covariance of the linear sequence $L_n$ can be obtained from the Lyapunov equation. Specifically, let $\Gamma$ be the unique solution of

$$A\Gamma + \Gamma A = S,$$

which can also be written as

$$\Gamma = \int_{\mathbb{R}^+} e^{-At} S e^{-At} dt. \tag{6}$$

Then the following limitation holds

$$\Gamma = \lim_{\substack{n\gamma\to\infty \\ \gamma\to 0}} \frac{\text{Cov}(L_n)}{\gamma}.$$

This asymptotic covariance is in accordance with results in [Pflug, 1986; Chen et al., 2022; Wei et al., 2025]. In the theorem below, we present a refined CLT result via the linear approximation technique.

**Theorem 3.4** (Quenched Central Limit Theorem). *Under same conditions of Lemma 3.3, let $n \to \infty$ and $\gamma \to 0$ such that $n\gamma \geq \nu \log n$ for some constant $\nu > 1/2\lambda$. Then, for an SGD sequence $\{\theta_n\}_{n \in \mathbb{N}}$ with arbitrarily initialization $\theta_0 \in \mathbb{R}^d$, we have*

$$\frac{\theta_n - \theta^*}{\sqrt{\gamma}} \xrightarrow{D} \mathcal{N}(0, \Gamma).$$

**Remark** (Quenched version of asymptotic normality). *In most classical CLT-type results for SGD, one assumes that the iterate sequence $\{\theta_n\}_{n \in \mathbb{N}}$ is stationary, which means choosing the initialization $\theta_0$ exactly following the limiting stationary distribution of $\theta_n$ [Pflug, 1986; Dieuleveut et al., 2020; Chen et al., 2022]. Instead, our quenched CLT yields a stronger result which guarantees the asymptotic normality of SGD sequences with any arbitrarily initial point $\theta_0 \in \mathbb{R}^d$.*

Notably, the relationship $n\gamma \gtrsim \log n$ is a minimal condition for SGD in most settings as discussed before. Consider the decaying learning rate schedule $\eta_n \asymp n^{-\alpha}$, taking $\alpha = 1$, the minimal rate of the total step size $\sum_{i=1}^{n} \eta_i$ is also $\mathcal{O}(\log n)$.

### 3.3 Non-asymptotic Gaussian approximation

The central limit theorem 3.4 establishes a weak convergence of SGD to the Gaussian distribution. However, it does not tell how close the SGD iterates are to Gaussian, and how fast it converges when the number of iterations grows or the learning rate is turned down. For example, it is natural to ask how small the following distance could be

$$|\mathbb{P}((\theta_n - \theta^*)/\sqrt{\gamma} \in \hat{\mathcal{C}}) - \mathbb{P}(\mathcal{N}(0, \Gamma) \in \hat{\mathcal{C}})|$$

for some confidence interval $\hat{\mathcal{C}}$ of concern, but the CLT can not help due to its asymptotic essence. Consequently, the effectiveness of statistical inference is still questionable with a finite sample and fixed learning rate.

This restriction motivates us to investigate the non-asymptotic convergence rate to normality. The Berry-Esseen theorem is a powerful tool to quantify the maximum approximation error. In the original groundbreaking work [Berry, 1941; Esseen, 1942], the Kolmogorov–Smirnov distance between the empirical distribution and the Gaussian is specified as

$$\sup_{x \in \mathbb{R}} |F_n(x) - \Phi(x)| \leq \frac{0.33554\mathbb{E}|X_1^3| + 0.4748(\mathbb{E}X_1^2)^{2/3}}{(\mathbb{E}X_1^2)^{2/3}\sqrt{n}},$$

where $F_n$ is the cumulative distribution function (cdf) of i.i.d. average of $X_1, ..., X_n$ with a finite third moment, and $\Phi$ is the standard normal cdf.

Taking advantage of the linear recursion and functional dependence measure, we develop the following optimal Gaussian approximation result for constant learning rate SGD.

**Theorem 3.5** (Berry-Esseen bound). *Suppose that Assumptions 2.1 and 2.2 hold with $p \geq 4$, and the same conditions of Lemma 3.3 hold. Let $\mathcal{V} = \{\mathcal{D} \in \mathbb{R}^d : \mathcal{D} \text{ is convex.}\}$ and $Y_\Gamma$ be a mean zero normal vector in $\mathbb{R}^d$ with covariance $\Gamma$. Then, we have the following Berry-Esseen inequality:*

$$\sup_{\mathcal{D} \in \mathcal{V}} |\mathbb{P}(\frac{\Delta_n}{\sqrt{\gamma}} \in \mathcal{D}) - \mathbb{P}(Y_\Gamma \in \mathcal{D})|$$

$$\leq C\big[\sqrt{\gamma} + (1 - \lambda\gamma)^{\frac{n-1}{2}} + \sqrt{\gamma}n(1 - \lambda\gamma)^{n-1} + \gamma + (1 - \lambda\gamma)^{2n}\big],$$

*where $C$ is a constant independent of $n$, $\gamma$ and $\theta_0$.*

**Remark** (Rate of Gaussian approximation). *The first term comes from the third moment of linear approximation sequence. The second and third terms come from the dependence structure of SGD and the error of linear approximation. The last two terms are due to the difference between finite-sample and asymptotic covariance. To the best of our knowledge, Theorem 3.5 is the first Gaussian approximation result for constant learning-rate SGD that explicitly bounds the distance between distributions with a specific order of $\gamma$ and $n$. The dominant term $\sqrt{\gamma}$ can not be improved, since the bias of SGD is $\mathcal{O}(\gamma)$ which can not be eliminated [Dieuleveut et al., 2020]. As a result, $\Delta_n/\sqrt{\gamma}$ is at least $\mathcal{O}(\sqrt{\gamma})$ away from the centered Gaussian vector.*

Suppose we know the number of iterations $n$ a priori. To choose an appropriate $\gamma$ scaling with $n$, the right-hand side can be nearly optimized by setting $\gamma \propto \nu \log n/n$ for some $\nu > 1/\lambda$. Then the order of Gaussian approximation becomes

$$\sup_{\mathcal{D} \in \mathcal{V}} |\mathbb{P}(\frac{\Delta_n}{\sqrt{\gamma}} \in \mathcal{D}) - \mathbb{P}(Y_\Gamma \in \mathcal{D})| \lesssim \frac{\sqrt{\log n}}{\sqrt{n}}.$$

A similar result of ASGD with decaying learning rate schedule $\eta_n \asymp n^{-\alpha}$ is developed by [Shao and Zhang, 2022; Samsonov et al., 2024], where the optimal approximation rate is $\mathcal{O}(n^{-1/4})$. In comparison, the normal approximation rate is sharper with a constant learning rate SGD.

## 4 Sharp concentration of tail probability

In addition to $L_p$ convergence, CLT, and Gaussian approximation, another vital concern is the tail behavior of the estimation error, as the expected error rate can not be achieved by a one-time implementation. Practitioners want to ensure an ideal performance of the algorithm in a single trial, especially under non-linear and heavy-tailed noisy settings. To this end, we generalize the method developed in Nagaev [1979] to provide a high probability guarantee of the estimation accuracy without the requirement of bounded or sub-Gaussian gradient noise.

### 4.1 Non-linear autoregressive approximation

We use the following non-linear autoregressive sequence to approximate the SGD sequence. Let $\beta_0 = \theta^*$ and
$$\beta_n = \beta_{n-1} - \gamma \nabla G(\beta_{n-1}) - \gamma \nabla g(\theta^*, X_n).$$
The sequence $\beta_n$ can be viewed as the gradient descent iteration with i.i.d. noise. The following lemma characterizes the convergence of $\beta_n$ and the approximation error rate.

**Lemma 4.1.** *Under same conditions of Theorem 2.3,*

$$\|\beta_n - \theta^*\|_p^2 \leq \frac{3(p-1)M_p^2}{\alpha_0}\gamma,$$

*and for some constant $\alpha_1$ such that*

$$\alpha_1 \leq \min\{2\mu - \gamma L_p^2, \frac{1}{\gamma}\},$$

*we have*

$$\|\theta_n - \beta_n\|_p^2 \leq [(1 - \alpha_1\gamma)^n + 4(p-1)L_p^2 n\gamma^2(1 - \alpha_0\gamma)^n]\rho_p^2 + \frac{3(p-1)M_p^2}{\alpha_0\alpha_1}\gamma^2.$$

**Remark.** *Compared to Lemma 3.3, the non-linear autoregression sequence can approximate the SGD in $L_p$-space for $p \geq 2$, which is more useful for a precise derivation of tail probability.*

### 4.2 Nagaev-type inequality of tail probability

To analyze the concentration property of SGD with fixed learning rates, we first focus on the linear functional of the estimation error, i.e., $v^\top(\theta_n - \theta^*)$ for all $v \in \mathbb{S}^d$, the unit sphere in $\mathbb{R}^d$. The tail probability bounds directly from the $L_p$ convergence or Gaussian approximation rate are too conservative. Simply applying the Berry-Esseen inequality 3.5, for instance, will result in a sub-Gaussian tail plus an inevitable $\sqrt{\gamma}$ term. Another naive way is to use the Markov inequality on Theorem 2.3. If we set the degree of tolerance as $\epsilon$, it yields

$$\mathbb{P}(|v^\top(\theta_n - \theta^*)| > \epsilon) \lesssim \frac{1}{\epsilon^p}[(1 - \alpha_0\gamma)^{np/2}\rho_p^p + M_p^p\alpha_0^{-p/2}\gamma^{p/2}], \tag{7}$$

where the polynomial term $\gamma^{p/2}$ is far from optimal. As a result, for some credible level $0 < \delta < 1$, with probability $1 - \delta$, one can only have

$$|v^\top(\theta_n - \theta^*)| = \mathcal{O}(\delta^{-1/p}\sqrt{\gamma}).$$

According to $n \gtrsim \gamma^{-1}$, the resulting estimate of sample complexity $N(\epsilon, \delta)$ (i.e., the minimum number of iterations such that the MSE satisfies the given credible level $\delta$ and degree of tolerance $\epsilon$),

$$N(\epsilon, \delta) = \mathcal{O}(\frac{\delta^{-2/p}}{\epsilon}),$$

grows rapidly for any tolerance level $\delta \in (0, 1)$.

The next theorem provides a Nagaev-type high-confidence bound that is substantially tighter than existing results by incorporating functional dependence measures, non-linear autoregressive approximation and the moment-generating function.

**Theorem 4.2** (Nagaev inequality). *Under same conditions of Theorem 2.3, for any $\epsilon > 0$, we have*

$$\sup_{v \in \mathbb{S}^d} \mathbb{P}(|\langle v, \Delta_n \rangle| > \epsilon) \leq \frac{C_1 \gamma^{p-1}}{\epsilon^p} + 2 \exp\{-\frac{C_2 \epsilon^2}{\gamma}\} + \frac{2^{p-1} \rho_p^p (1 - \alpha_0 \gamma)^{np/2}}{\epsilon^p} \left(1 + 4(p-1)L_p^2 n \gamma^2\right)^{p/2},$$

*where $C_1$ and $C_2$ are some positive constants independent of $\epsilon$, $n$, $\gamma$ and $\theta_0$.*

**Remark** (Rate of tail probability). *The polynomial term in Theorem 4.2 is $\gamma^{p-1}$, much sharper than those obtained from Gaussian approximation or Markov inequality when $p > 2$. The sub-Gaussian term $\exp\{-C_2 \epsilon^2 / \gamma\}$ is optimal in the sense that $\Delta_n / \sqrt{\gamma}$ is asymptotically normal. With a high degree of tolerance $\epsilon$, the polynomial term dominates, and the estimate of sample complexity can be greatly improved to $\mathcal{O}(\delta^{-1/(p-1)} \epsilon^{-p/2(p-1)})$ in this case. With a low degree of tolerance, the sub-Gaussian term dominates.*

We can directly obtain the same result of the uniform distance between $\theta_n$ and $\theta^*$ by the union bound. Taking $v$ over the standard basis $(0, ..., 0, 1, 0, ..., 0)^\top$ yields

$$\mathbb{P}(\|\Delta_n\|_\infty > \epsilon) \leq \frac{C_1 d \gamma^{p-1}}{\epsilon^p} + 2d \exp\{-\frac{C_2 \epsilon^2}{\gamma}\}$$
$$+ \frac{2^{p-1} d \rho_p^p (1 - \alpha_0 \gamma)^{np/2}}{\epsilon^p} [1 + 4(p-1) L_p^2 n \gamma^2]^{p/2}.$$

# 5 Numerical studies

## 5.1 Simulation setting

We conduct a simulation to demonstrate that our Nagaev-type inequality in Theorem 4.2 is indeed valid and tight. Consider the following data generating mechanism for the logistic regression model: $X_i = (a_i, b_i), i = 1, 2, ...$ are i.i.d. random vectors where $a_i$ are generated from a 5-dimensional independent t distribution with degrees of freedom df $= 3$. $b_i \in \{1, -1\}$ follows a Bernoulli distribution with the probability given by $\mathbb{P}(b_i | a_i) = 1/(1 + \exp(-b_i a_i^\top \theta^*))$. The loss function is defined as the negative log-likelihood,

$$g(\theta, X_i) = \log(1 + \exp(-b_i a_i^\top \theta)).$$

We investigate logistic regression for its non-linearity. The t distribution with df $= \nu$ only has finite $p$-th moments with $p < \nu$. This property enables us to study the performance of different tail probability bounds with specific values of $p$. Elementary calculation shows that

$$\nabla g(\theta, X) = \nabla g(\theta, a, b) = \frac{-ba}{1 + \exp(ba^\top \theta)},$$

and therefore $\nabla g(\theta^*, X)$ only has finite $p$-th moment with $p < 3$ in our setting. Theoretically, we can choose $p$ arbitrarily close to 3 and apply the inequalities discussed in Section 4.2. For simplicity, we take $p = 3$ since the results we report in the simulation are continuous with respect to $p$.

We run 1000 independent trials with $n = 500000$ and $\gamma = 0.005, 0.001, 0.0002$. Since the number of iterations is large enough, the main contribution in the tail probability bounds are the polynomial terms and the sub-Gaussian term, i.e.,

$$II_1 = \frac{\gamma^{p/2}}{\epsilon^p}$$

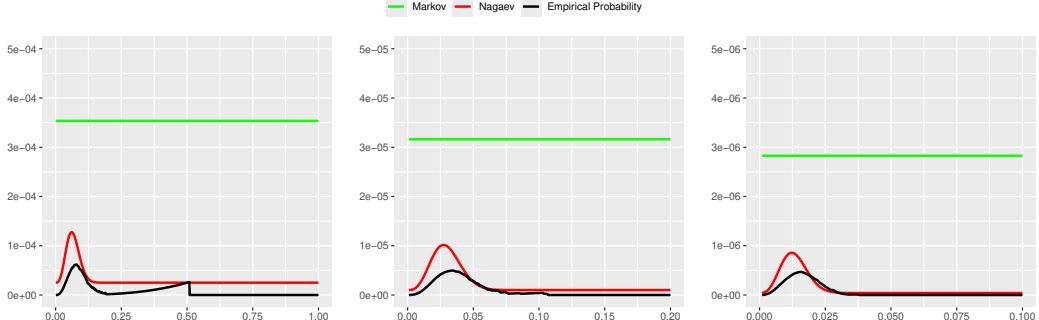

Figure 1: Comparison of different tail probability bounds against the empirical probability. The learning rate $\gamma = 0.005, 0.001, 0.0002$ from left to right.

in the Markov-type bound, and

$$II_2 = \frac{\gamma^{p-1}}{\epsilon^p} + 2\exp\{-\frac{C_2\epsilon^2}{\gamma}\}$$

in the Nagaev-type bound. In our simulation, we set $C_2 = 2$, because Theorem 3.4 provides the equation that the asymptotic covariance $\Gamma$ satisfies. Elementary calculation Chen et al. [2020] shows that for the logistic model, $A = S$, and the unique solution to $A\Gamma + \Gamma A = S$ becomes $\Gamma = \mathbf{I}_d/2$, so we accordingly scale the sub-Gaussian term.

## 5.2 Numerical results

We compare the concentration inequality (7) and our Nagaev-type bound, Theorem 4.2, both with the empirical probability: $II_3 = \text{Avg}_v(\mathbb{P}(|v^\top(\theta_n - \theta^*)| > \epsilon))$, where $v$ ranges over the standard basis vectors $(0, \ldots, 0, 1, 0, \ldots, 0)^\top$. For more transparent visualization, we multiply these quantities by $\epsilon^p$ and plot the results against the degree of tolerance $\epsilon$. In Figure 1, the x-axis is the degree of tolerance. The green, red, and black curves are $\epsilon^p II_1$, $\epsilon^p II_2$, and $\epsilon^p II_3$, respectively. They represent the $\epsilon^p$-scaled Markov bound, Nagaev bound, and empirical probability.

Figure 1 clearly indicates that the tail probability bound from Markov inequality is excessively conservative. In contrast, our Nagaev-type inequality from Theorem 4.2 yields a much sharper upper bound for SGD across different learning-rate scales. The shape of empirical probability closely matches the dichotomous phenomenon in theory: the dominance transits from the polynomial term to the sub-Gaussian term as $\epsilon$ decreases. The experiment results confirm that our Nagaev-type bound is both valid and tight, precisely describing the tail behavior of constant learning rate SGD.

## Acknowledgments and Disclosure of Funding

We sincerely thank the program chair, senior area chair, area chair, and the five reviewers for their constructive feedback and involved discussion, which has greatly improved the clarity of our paper. Jiaqi Li's research is partially supported by the NSF (Grant NSF/DMS-2515926). Wei Biao Wu's research is partially supported by the NSF (Grant NSF/DMS-2311249). We would like to thank Johannes Schmidt-Hieber for helpful discussions.

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

# A  Technical Appendices and Supplementary Material

All theoretical results are proved in the appendix. We first introduce some technical lemma.

**Lemma A.1** (Rio's inequality [Rio, 2009])**.** *Let $X \in \mathbb{R}^d$ and $Y \in \mathbb{R}^d$ be two random vectors such that $\mathbb{E}|X|^p < \infty$ and $\mathbb{E}|Y|^p < \infty$ for some $p \geq 2$. Then we have*

$$\|X + Y\|_p^2 \leq \|X\|_p^2 + (p-1)\|Y\|_p^2. \tag{8}$$

**Lemma A.2.** *Under Assumption 2.1 and 2.2, we have*

$$|\nabla G(\theta_1) - \nabla G(\theta_2)| \leq L_p|\theta_1 - \theta_2|.$$

*Proof.* By Assumption 2.2,

$$\|\nabla g(\theta_1, X) - \nabla g(\theta_2, X)\|_2^2 \leq L_p^2|\theta_1 - \theta_2|^2, \quad \text{for all } \theta_1, \theta_2 \in \mathbb{R}^d.$$

By the convexity of $|\cdot|^2$ and Jensen's inequality,

$$\begin{aligned}
|\nabla G(\theta_1) - \nabla G(\theta_2)|^2 &= |\mathbb{E}_X(\nabla g(\theta_1, X) - \nabla g(\theta_2, X))|^2 \\
&\leq \mathbb{E}_X|\nabla g(\theta_1, X) - \nabla g(\theta_2, X)|^2 \\
&\leq L_p^2|\theta_1 - \theta_2|^2.
\end{aligned}$$

$\square$

## A.1  Proof of Moments Convergence of SGD

**Lemma A.3.** *Consider the SGD iterates $\{\theta_t\}_{t \geq 1}$ in (2). Under the same conditions of Theorem 2.3, for some universal constant $\alpha_0$ such that*

$$0 < \alpha_0 \leq \min\left\{\frac{1}{\gamma}, 2\mu - (6p - 5)L_p^2\gamma\right\}, \tag{9}$$

*we have, for all $n \geq 1$,*

$$\|\Delta_n\|_p^2 \leq (1 - \alpha_0\gamma)\|\Delta_{n-1}\|_p^2 + 3(p-1)\gamma^2 M_p^2. \tag{10}$$

*Proof of Lemma A.3.* Since $\xi_t$, for $t \geq 1$, are i.i.d. random samples, it follows from the tower rule that

$$\mathbb{E}[\nabla g(\theta_{n-1}, \xi_n) - \nabla G(\theta_{n-1}) \mid \theta_{n-1}] = 0. \tag{11}$$

Therefore, by applying Rio's inequality in Lemma A.1, for $p \geq 2$, we have

$$\begin{aligned}
\|\Delta_n\|_p^2 &\leq \|\theta_{n-1} - \theta^* - \gamma\nabla G(\theta_{n-1})\|_p^2 + (p-1)\gamma^2\|\nabla g(\theta_{n-1}, \xi_n) - \nabla G(\theta_{n-1})\|_p^2 \\
&=: \mathbb{I}_1 + \mathbb{I}_2.
\end{aligned} \tag{12}$$

We shall bound the two parts $\mathbb{I}_1$ and $\mathbb{I}_2$ separately. For the first part $\mathbb{I}_1$, note that $\nabla G(\theta^*) = 0$ and by the triangle inequality, we have

$$\begin{aligned}
\mathbb{I}_1 &= \|\theta_{n-1} - \theta^* - \gamma\nabla G(\theta_{n-1})\|_p^2 \\
&= \Big\| \langle\theta_{n-1} - \theta^*, \theta_{n-1} - \theta^*\rangle - 2\gamma\langle\theta_{n-1} - \theta^*, \nabla G(\theta_{n-1}) - \nabla G(\theta^*)\rangle \\
&\quad + \gamma^2\langle\nabla G(\theta_{n-1}) - \nabla G(\theta^*), \nabla G(\theta_{n-1}) - \nabla G(\theta^*)\rangle \Big\|_{p/2} \\
&\leq \Big\| \langle\theta_{n-1} - \theta^*, \theta_{n-1} - \theta^*\rangle - 2\gamma\langle\theta_{n-1} - \theta^*, \nabla G(\theta_{n-1}) - \nabla G(\theta^*)\rangle \Big\|_{p/2} \\
&\quad + \gamma^2\big\|\nabla G(\theta_{n-1}) - \nabla G(\theta^*)\big\|_p^2.
\end{aligned} \tag{13}$$

By applying Assumption 2.1 to the first term and Assumption 2.2 to the second term, we can obtain

$$\mathbb{I}_1 \leq (1 - 2\gamma\mu + \gamma^2 L_p^2)\|\theta_{n-1} - \theta^*\|_p^2. \tag{14}$$

Regarding the second part $\mathbb{I}_2$, since $\nabla G(\theta^*) = 0$, we have

$$\begin{aligned}
&\|\nabla g(\theta_{n-1}, \xi_n) - \nabla G(\theta_{n-1})\|_p \\
&\leq \|\nabla g(\theta_{n-1}, \xi_n) - \nabla g(\theta^*, \xi_n)\|_p + \|\nabla G(\theta_{n-1}) - \nabla G(\theta^*)\|_p + \|\nabla g(\theta^*, \xi_n)\|_p.
\end{aligned} \quad (15)$$

Hence, by Assumption 2.2, we can achieve

$$\|\nabla g(\theta_{n-1}, \xi_n) - \nabla G(\theta_{n-1})\|_p^2 \leq 6L_p^2 \|\theta_{n-1} - \theta^*\|_p^2 + 3\|\nabla g(\theta^*, \xi_n)\|_p^2. \quad (16)$$

Combining results from $\mathbb{I}_1$ and $\mathbb{I}_2$, we can obtain

$$\|\Delta_n\|_p^2 \leq (1 - 2\gamma\mu + (6p-5)\gamma^2 L_p^2)\|\theta_{n-1} - \theta^*\|_p^2 + 3(p-1)\gamma^2\|\nabla g(\theta^*, \xi_n)\|_p^2.$$

This can directly lead to the desired inequality. $\qquad \square$

*Proof of Theorem 2.3.* By recursively applying Lemma A.3, we have

$$\|\Delta_n\|_p^2 \leq \prod_{k=1}^{n}(1 - \alpha_0\gamma)\|\Delta_0\|_p^2 + 3(p-1)M_p^2 \sum_{j=1}^{n}\gamma^2 \prod_{k=j+1}^{n}(1 - \alpha_0\gamma). \quad (17)$$

By elementary calculations,

$$\|\Delta_n\|_p^2 \leq (1 - \alpha_0\gamma)^n \rho_p^2 + 3(p-1)M_p^2\gamma\alpha_0^{-1}. \quad (18)$$

$$\square$$

## A.2 Proof of the Bound for Functional Dependence Measure

Recall that the SGD sequence $\theta_n$ can be represented by $\theta_n = \tau_n(X_1, ..., X_n)$ and $\theta_n^{(t)} = \tau_n(X_1, ..., X_{t-1}, X_t', X_{t+1}, ..., X_n)$ for some measurable function $\tau_n$ that can vary for different $n$, where $X_t'$ is an i.i.d. copy of $X_t$. The functional dependence measure was defined as $\psi(n, t, p) = \|\theta_n - \theta_n^{(t)}\|_p$. We prove the bound for $\psi(n, t, p)$ as stated in Theorem 3.1, which is fundamental to the proofs of CLT, Berry-Esseen inequality and Nagaev-type inequality.

*Proof.* By applying Rio's inequality, for each $t \leq n - 1$, we have

$$\begin{aligned}
\|\theta_n - \theta_n^{(t)}\|_p^2 &\leq (1 - 2\gamma\mu + (6p-5)\gamma^2 L_p^2)^{n-t}\|\theta_t - \theta_t^{(t)}\|_p^2 \\
&\leq (1 - \alpha_0\gamma)^{n-t}\|\theta_t - \theta_t^{(t)}\|_p^2.
\end{aligned} \quad (19)$$

It follows from Assumption 2.2 that for all $t \geq 1$,

$$\begin{aligned}
\|\nabla g(\theta_{t-1}, X_t)\|_p^2 &\leq 2\|\nabla g(\theta_{t-1}, X_t) - \nabla g(\theta^*, X_t)\|_p^2 + 2\|\nabla g(\theta^*, X_t)\|_p^2 \\
&\leq 2L_p^2\|\theta_{t-1} - \theta^*\|_p^2 + 2M_p^2.
\end{aligned} \quad (20)$$

As a direct consequence, we can achieve

$$\begin{aligned}
\|\theta_t - \theta_t^{(t)}\|_p^2 &= \gamma^2\|\nabla g(\theta_{t-1}, X_t) - \nabla g(\theta_{t-1}, X_t')\|_p^2 \\
&\leq \gamma^2\left(2\|\nabla g(\theta_{t-1}, X_t)\|_p^2 + 2\|\nabla g(\theta_{t-1}, X_t')\|_p^2\right) \\
&\leq 4\gamma^2\left(L_p^2\|\theta_{t-1} - \theta^*\|_p^2 + M_p^2\right).
\end{aligned} \quad (21)$$

This along with expression (19) and Theorem 2.3 provides the desired result. $\qquad \square$

## A.3 Proof of Quenched CLT

Here and in the sequel, we will repeatedly use a basic property that $|\mathbf{I}_d - \gamma A| \leq 1 - \gamma\lambda_* \leq 1 - \gamma\lambda$. We denote $Z_i = \nabla g(\theta^*, X_i)$ as the gradient noise.

*Proof of Lemma 3.3.* We first show that $I_{1,n}$ and $I_{2,n}$ vanish. Define $\mathcal{F}_0 = \emptyset$ and $\mathcal{F}_t = \sigma(X_1, ..., X_t)$ as the filtration generated by the data. It is clear that $\{D_n\}$ is a martingale difference sequence w.r.t. the filtration $\{\mathcal{F}_t\}$, and we have

$$\|I_{2,n}\|^2 \leq \gamma^2 \sum_{k=1}^{n} \|(\mathbf{I}_d - \gamma A)\|^{2n-2k} \|D_k\|^2$$

$$\leq 4\gamma^2 L_2^2 \sum_{k=1}^{n} \|(\mathbf{I}_d - \gamma A)\|^{2n-2k} \|\theta_{k-1} - \theta^*\|^2$$

$$\leq 4\gamma^2 L_2^2 \sum_{k=1}^{n} \|(\mathbf{I}_d - \gamma A)\|^{2n-2k} (1 - \alpha_0\gamma)^{k-1} \rho_2^2 + 12 L_2^2 \gamma^3 M_2^2 \alpha_0^{-1} \sum_{k=1}^{n} \|(\mathbf{I}_d - \gamma A)\|^{2n-2k}$$

$$\leq 4\gamma^2 L_2^2 \rho_2^2 \frac{(1 - \gamma\lambda)^{n-1} - (1 - \gamma\lambda)^{2n-1}}{\gamma\lambda} + 12 L_2^2 \gamma^3 M_2^2 \alpha_0^{-1} \frac{1 - (1 - \gamma\lambda)^{2n}}{2\gamma\lambda - \gamma^2\lambda^2}$$

$$\leq 4 L_2^2 \rho_2^2 \gamma \frac{(1 - \lambda\gamma)^{n-1}}{\lambda} + \frac{12 L_2^2 \gamma^2 M_2^2}{\alpha_0(2\lambda - \gamma\lambda^2)}$$

$$\leq 4 L_2^2 \rho_2^2 \gamma \frac{(1 - \lambda\gamma)^{n-1}}{\lambda} + \frac{12 L_2^2 \gamma^2 M_2^2}{\alpha_0\lambda}.$$

Here we use the fact that $|D_n| \leq 2L_2|\Delta_{n-1}|$ due to Assumption 2.2, Lemma A.2, and the triangular inequality. The last inequality comes from $\gamma \leq 1/\alpha_0$ and $\gamma \leq 1/\lambda_*$. By Taylor expansion around $\theta^*$, since $\nabla G(\theta^*) = 0$, we have

$$R_n = A\Delta_{n-1} - (\nabla G(\theta_{n-1}) - \nabla G(\theta^*))$$

$$= -\int_0^1 [\nabla^2 G(\theta^* + t(\theta_{n-1} - \theta^*)) - \nabla^2 G(\theta^*)](\theta_{n-1} - \theta^*) dt.$$

By Assumption 3.2, when $|\theta_{n-1} - \theta^*| \leq \kappa$, we have $|R_n| \leq L|\theta_{n-1} - \theta^*|^2$. For $|\theta_{n-1} - \theta^*| > \kappa$, the Lipschitz continuity of the gradient implies $|R_n| \leq 2L_2|\theta_{n-1} - \theta^*| \leq 2L_2\kappa^{-1}|\theta_{n-1} - \theta^*|^2$. So we finally have $|R_n| \leq C_0|\Delta_{n-1}|^2$ where $C_0 = \max\{L, 2L_2\kappa^{-1}\}$. As a result,

$$\mathbb{E}|I_{1,n}| \leq C_0\gamma \sum_{k=1}^{n} \|\mathbf{I}_d - \gamma A\|^{n-k} \|\theta_{k-1} - \theta^*\|^2$$

$$\leq n\gamma C_0 \rho_2^2 (1 - \gamma\lambda)^{n-1} + 3C_0 M_2^2 \alpha_0^{-1} \gamma^2 \frac{1 - (1 - \gamma\lambda)^n}{\gamma\lambda}$$

$$\leq n\gamma C_0 \rho_2^2 (1 - \gamma\lambda)^{n-1} + \frac{3C_0 M_2^2}{\lambda\alpha_0}\gamma. \tag{22}$$

We also have $\|(\mathbf{I}_d - \gamma A)^n \Delta_0\| \leq (1 - \gamma_*\lambda)^n \rho_1$. Combining these inequalities will lead to the bound in Lemma 3.3. $\qquad\square$

*Proof of Theorem 3.4.* As $\gamma \to 0$ and $n\gamma \geq \nu \log n$ for some constant $\nu > 1/2\lambda$, elementary calculation shows that the difference between $\Delta_n/\sqrt{\gamma}$ and $L_n/\sqrt{\gamma}$ goes to 0, i.e., by Lemma 3.3 we have $\|\Delta_n/\sqrt{\gamma} - L_n/\sqrt{\gamma}\|_1 \to 0$. Then it suffices to prove that

$$\frac{L_n}{\sqrt{\gamma}} \xrightarrow{D} \mathcal{N}(0, \Gamma). \tag{23}$$

Notice that $L_n/\sqrt{\gamma}$ is a linear combination of i.i.d. random vectors, the covariance matrix of which is

$$\Gamma_n(\gamma) = \gamma \sum_{k=1}^{n} (\mathbf{I}_d - \gamma A)^{n-k} S(\mathbf{I}_d - \gamma A)^{n-k}.$$

We introduce the following auxiliary lemma.

**Lemma A.4.** *The minimum eigenvalue of $\Gamma_n(\gamma)$, denoted as $\lambda_n(\gamma)$, satisfies*

$$\lambda_n(\gamma) \geq \frac{\lambda_S(1 - (1 - \gamma\lambda^*)^{2n})}{2\lambda^*},$$

*where $\lambda_S$ is the smallest eigenvalue of $S$.*

This lemma is easy to prove since $(\mathbf{I}_d - \gamma A)$ is positive definite with the minimum eigenvalue $1 - \gamma \lambda^*$. So we have

$$\lambda_n(\gamma) \geq \lambda \sum_{k=1}^{n} (1 - \gamma \lambda^*)^{2n-2k} \lambda_S$$

and elementary calculation leads to the conclusion. The lemma implies that $|\Gamma_n(\gamma)^{-1}|$ is bounded by some constant. As $\gamma \to 0$ and $n\gamma \to \infty$, we have

$$\max_{1 \leq k \leq n} |\Gamma_n(\gamma)^{-1} \gamma (\mathbf{I}_d - \gamma A)^{n-k} S (\mathbf{I}_d - \gamma A)^{n-k}| \lesssim \gamma \to 0.$$

By the multivariate Lindeberg-Feller CLT, it is clear that

$$\Gamma_n(\gamma)^{-1/2} \frac{L_n}{\sqrt{\gamma}} \xrightarrow{D} \mathcal{N}(0, \mathbf{I}_d). \tag{24}$$

To determine the closeness between $\Gamma_n(\gamma)$ and $\Gamma$, notice that

$$\Gamma_{n+1}(\gamma) = (\mathbf{I}_d - \gamma A)\Gamma_n(\gamma)(\mathbf{I}_d - \gamma A) + \gamma S.$$

Minus $\Gamma$ on both sides and plug $A\Gamma + \Gamma A = S$ into the formula above, we get

$$\Gamma_{n+1}(\gamma) - \Gamma = (\mathbf{I}_d - \gamma A)(\Gamma_n(\gamma) - \Gamma)(\mathbf{I}_d - \gamma A) + \gamma^2 A\Gamma A.$$

Hence, there exists a universal constant $C$ such that

$$|\Gamma_{n+1}(\gamma) - \Gamma| \leq (1 - \gamma\lambda)^2 |\Gamma_n(\gamma) - \Gamma| + C\gamma^2.$$

Let $\Gamma_0(\gamma) = \mathbf{0}_{d \times d}$. Recursively updating the inequality we get

$$|\Gamma_n(\gamma) - \Gamma| \leq \sum_{i=1}^{n} C\gamma^2 (1 - \gamma\lambda)^{2(n-i)} + (1 - \gamma\lambda)^{2n}|\Gamma_0(\gamma) - \Gamma| \tag{25}$$

$$\leq \frac{C\gamma}{2\lambda - \gamma\lambda^2} + (1 - \gamma\lambda)^{2n}|\Gamma| \to 0. \tag{26}$$

Since the eigenvalues of $\Gamma_n(\gamma)$ are bounded and bounded away from 0, and $\Gamma$ is a fixed positive definite matrix, we have $\Gamma^{-1/2}\Gamma_n(\gamma)^{1/2} \to \mathbf{I}_d$. By Slustky's theorem,

$$\Gamma^{-1/2} \frac{L_n}{\sqrt{\gamma}} = \Gamma^{-1/2}\Gamma_n(\gamma)^{1/2}\Gamma_n(\gamma)^{-1/2} \frac{L_n}{\sqrt{\gamma}} \xrightarrow{D} \mathcal{N}(0, \mathbf{I}_d),$$

and we proved (23). $\qquad\square$

**Remark on the multivariate Lindeberg–Feller CLT**

Suppose a triangular sequence $y_{n,k} \in \mathbb{R}^d$ are independent with means $\mathbb{E}y_{n,k} = 0$ and covariance matrices $V_{n,k} = \mathbb{E}(y_{n,k}y_{n,k}^\top)$. Set

$$U_n = \sum_{k=1}^{n} V_{n,k}, \quad \nu_n^2 = \lambda_{\min}(U_n).$$

If $\nu_n^2 > 0$ and for all $\varepsilon > 0$

$$\lim_{n \to \infty} \frac{1}{\nu_n^2} \sum_{k=1}^{n} \mathbb{E}\big(|y_{n,k}|^2 \cdot \mathbf{1}(|y_{n,k}|^2 > \varepsilon\nu_n^2)\big) = 0,$$

then as $n \to \infty$

$$U_n^{-1/2} \left(\sum_{k=1}^{n} y_{n,k}\right) \Rightarrow \mathcal{N}(0, I_d).$$

Above is the statement of the multivariate Lindeberg–Feller CLT. We apply the theorem to the standardized sum

$$\Gamma_n(\gamma)^{-1/2}\sqrt{\gamma} \sum_{k=1}^{n} (I_d - \gamma A)^{n-k} Z_k.$$

Denote

$$B_{n,k}(\gamma) = \Gamma_n(\gamma)^{-1/2}\sqrt{\gamma}(I_d - \gamma A)^{n-k}.$$

Then

$$y_{n,k} = B_{n,k}(\gamma)Z_k, \quad U_n = I_d, \quad \nu_n^2 = 1.$$

Let $M_n(\gamma)$ be the maximum norm of the matrix prefactor of $y_{n,k}$, i.e.

$$M_n(\gamma) = \max_{1 \le k \le n} |B_{n,k}(\gamma)| = \max_{1 \le k \le n} |\Gamma_n(\gamma)^{-1}\gamma(I_d - \gamma A)^{2n-2k}|.$$

We have shown that $|\Gamma_n(\gamma)^{-1}|$ is bounded by some constant. As a result, $M_n(\gamma) \lesssim \gamma \to 0$. This is sufficient for the Lindeberg condition and CLT. To this end, notice that

$$\sum_{k=1}^{n} \mathbb{E}\big(|y_{n,k}|^2\mathbf{1}(|y_{n,k}|^2 \ge \varepsilon)\big) \le \sum_{k=1}^{n} |B_{n,k}(\gamma)|^2\mathbb{E}\big(|Z_k|^2\mathbf{1}(M_n(\gamma)|Z_k|^2 \ge \varepsilon)\big).$$

By the dominated convergence theorem, we have

$$\mathbb{E}\big(|Z_k|^2\mathbf{1}(M_n(\gamma)|Z_k|^2 \ge \varepsilon)\big) \to 0$$

due to $\mathbf{1}(M_n(\gamma)|Z_k|^2 \ge \varepsilon) \to 0$. By elementary calculation, $\sum_{k=1}^{n} |B_{n,k}(\gamma)|^2$ is bounded. As a result, the Lindeberg condition is justified.

### A.4 Proof of the Berry-Esseen Inequality

We begin with a refined analysis of the linear approximation.

$$\Delta_n = (\mathbf{I}_d - \gamma A)^n \Delta_0 - L_n + I_{1,n} + I_{2,n},$$

where $I_{1,n} = \gamma \sum_{k=1}^{n}(\mathbf{I}_d - \gamma A)^{n-k}R_k$, $I_{2,n} = \gamma \sum_{k=1}^{n}(\mathbf{I}_d - \gamma A)^{n-k}D_k$, $R_n = A\Delta_{n-1} - \nabla G(\theta_{n-1})$, and $D_n = \nabla G(\theta_{n-1}) - \nabla g(\theta_{n-1}, X_n) + \nabla g(\theta^*, X_n)$. By Lemma A.4, the matrix $\Gamma_n(\gamma)$ is invertible, and

$$\sqrt{\gamma}\Gamma_n(\gamma)^{-1/2} \lesssim \sqrt{\gamma}.$$

We further have the following decomposition:

$$\Gamma^{-1/2}\frac{\Delta_n}{\sqrt{\gamma}} = (\Gamma^{-1/2} - \Gamma_n(\gamma)^{-1/2})\frac{\Delta_n}{\sqrt{\gamma}} + \Gamma_n(\gamma)^{-1/2}\frac{(\mathbf{I}_d - \gamma A)^n\Delta_0 - L_n + I_{1,n} + I_{2,n}}{\sqrt{\gamma}}.$$

The Gaussian approximation error will be assessed via those terms. Define

$$\mathcal{I}_n = \sqrt{\gamma}\sum_{k=1}^{n}(1 - \gamma\lambda)^{n-k}|\Delta_{k-1}|^2,$$

as an auxiliary sequence. Define $\mathcal{I}_n^{(t)}$, $D_k^{(t)}$, and $I_{2,n}^{(t)}$ in the same way as $\theta_n^{(t)}$, and

$$\psi_D(n, t) = \|I_{2,n} - I_{2,n}^{(t)}\|_2$$

as the functional dependence measure of $I_{2,n}$. The next Lemma investigates the behavior of $\psi_D$.

**Lemma A.5.** *For $1 \le t \le n$,*

$$\psi_D(n, t)^2 \lesssim \gamma^2(1 - \gamma\lambda)^{2n-t-1} + \gamma^3(1 - \gamma\lambda)^{n-t-1}$$

*Proof.* Notice that

$$D_k - D_k^{(t)} = \begin{cases} 0 & k < t; \\ -\nabla g(\theta_{t-1}, X_t) + \nabla g(\theta^*, X_t) + -\nabla g(\theta_{t-1}, X_t') - \nabla g(\theta^*, X_t') & k = t; \\ \nabla G(\theta_{k-1}) - \nabla g(\theta_{k-1}, X_k) - [\nabla G(\theta_{k-1}^{(t)}) - \nabla g(\theta_{k-1}^{(t)}, X_k)] & k > t, \end{cases}$$

it is clear that that $D_k - D_k^{(t)}$ is also a martingale difference sequence. Due to stochastic Lipschitz continuity,

$$
\psi_D(n,t)^2 = \gamma^2 \mathbb{E}|\sum_{k=1}^{n}(\mathbf{I}_d - \gamma A)^{n-k}(D_k - D_k^{(t)})|^2
$$

$$
= \gamma^2 \sum_{k=1}^{n}(\mathbf{I}_d - \gamma A)^{2n-2k}\mathbb{E}|D_k - D_k^{(t)}|^2
$$

$$
\leq \gamma^2(1-\gamma\lambda)^{2n-2t}\mathbb{E}|D_t - D_t^{(t)}|^2 + 4L_2^2\gamma^2 \sum_{k=t+1}^{n}(1-\gamma\lambda)^{2n-2k}\mathbb{E}|\theta_{k-1} - \theta_{k-1}^{(t)}|^2
$$

By Theorem 2.3 and 3.1, we have

$$
\mathbb{E}|D_t - D_t^{(t)}|^2 \leq 4\|\nabla g(\theta_{t-1}, X_t) - \nabla g(\theta^*, X_t)\|^2 \leq 4L_2^2\|\Delta_{t-1}\|^2 \lesssim \gamma + (1-\alpha_0\gamma)^{t-1},
$$

and

$$
\mathbb{E}|\theta_{k-1} - \theta_{k-1}^{(t)}|^2 = \psi(k-1,t)^2 \lesssim \gamma^2(1-\alpha_0\gamma)^{k-1-t}.
$$

By elementary calculation,

$$
\psi_D(n,t)^2 \lesssim \gamma^3(1-\gamma\lambda)^{2n-2t} + \gamma^2(1-\gamma\lambda)^{2n-t-1} + \gamma^3(1-\gamma\lambda)^{n-t-1}
$$
$$
\asymp \gamma^2(1-\gamma\lambda)^{2n-t-1} + \gamma^3(1-\gamma\lambda)^{n-t-1}.
$$

$\square$

Now we are ready to prove the Berry-Esseen inequality.

*Proof of Theorem 3.5.* We apply Theorem 2.1 in Shao and Zhang [2022]. Since we use $L_n/\sqrt{\gamma} = \sqrt{\gamma}\sum_{k=1}^{n}(\mathbf{I}_d - \gamma A)^{n-k}Z_k$ to approximate $\Delta_n/\sqrt{\gamma}$, with

$$
\|L_n\|_3^3 \leq \sum_{k=1}^{n}\|\sqrt{\gamma}(\mathbf{I}_d - \gamma A)^{n-k}Z_k\|_3^3 \lesssim \sum_{k=1}^{n}\gamma^{3/2}(1-\lambda\gamma)^{n-k} \lesssim \sqrt{\gamma},
$$

Theorem 2.1 in Shao and Zhang [2022] states that

$$
\sup_{\mathcal{D}\in\mathcal{V}}|\mathbb{P}(\Gamma_n(\gamma)^{-1/2}\frac{\Delta_n}{\sqrt{\gamma}} \in \mathcal{D}) - \mathbb{P}(\mathcal{N}(0,\mathbf{I}_d) \in \mathcal{D})|
$$

$$
\leq C\Big(\sqrt{\gamma} + \mathbb{E}\{|\Gamma_n(\gamma)^{-1/2}L_n/\sqrt{\gamma}|\mathcal{J}\} + \sum_{t=1}^{n}\mathbb{E}\{|\mathcal{J} - \mathcal{J}^{(t)}||\Gamma_n(\gamma)^{-1/2}\sqrt{\gamma}(\mathbf{I}_d - \gamma A)^{n-t}Z_t|\}\Big),
$$

$$(27)$$

where $\mathcal{J}$ and $\mathcal{J}^{(t)}$ are some quantities such that

$$
\mathcal{J} \geq |\Gamma_n(\gamma)^{-1/2}\frac{\Delta_n - L_n}{\sqrt{\gamma}}| \tag{28}
$$

and $\mathcal{J}^{(t)}$ is independent of $X_t$ or $Z_t$. We define

$$
\mathcal{J} = |\Gamma_n(\gamma)^{-1/2}\sqrt{\gamma}(\mathbf{I}_d - \gamma A)^n\Delta_0| + |\gamma^{-1/2}\Gamma_n(\gamma)^{-1/2}I_{2,n}| + \tilde{C}\mathcal{I}_n,
$$

with some universal constant $\tilde{C}$ such that $\tilde{C}\mathcal{I}_n \geq |\gamma^{-1/2}\Gamma_n(\gamma)^{-1/2}I_{1,n}|$. Such $\tilde{C}$ exists because of the construction of $\mathcal{I}_n$ and Taylor's expansion. Then (28) holds by the triangle inequality. We further define

$$
\mathcal{J}^{(t)} = |\Gamma_n(\gamma)^{-1/2}\sqrt{\gamma}(\mathbf{I}_d - \gamma A)^n\Delta_0| + |\gamma^{-1/2}\Gamma_n(\gamma)^{-1/2}I_{2,n}^{(t)}| + \tilde{C}\mathcal{I}_n^{(t)}
$$

such that $\mathcal{J}^{(t)}$ is independent of $X_t$ and $Z_t$, and decompose the difference between $\mathcal{J}$ and its copula perturbation $\mathcal{J}^{(t)}$ as

$$
|\mathcal{J} - \mathcal{J}^{(t)}| \leq \tilde{C}|\mathcal{I}_n - \mathcal{I}_n^{(t)}| + \Big||\gamma^{-1/2}\Gamma_n(\gamma)^{-1/2}I_{2,n}| - |\gamma^{-1/2}\Gamma_n(\gamma)^{-1/2}I_{2,n}^{(t)}|\Big|
$$

$$
\leq \tilde{C}|\mathcal{I}_n - \mathcal{I}_n^{(t)}| + |\gamma^{-1/2}\Gamma_n(\gamma)^{-1/2}(I_{2,n} - I_{2,n}^{(t)})|. \tag{29}
$$

We will control each error term decomposed above. We first investigate the following quantity:

$$\mathbb{E}|\mathcal{I}_n - \mathcal{I}_n^{(t)}||\Gamma_n(\gamma)^{-1/2}\sqrt{\gamma}(\mathbf{I}_d - \gamma A)^{n-t}Z_t|$$

$$\lesssim \gamma(1-\gamma\lambda)^{n-t}\mathbb{E}\{|Z_t|\sum_{k=1}^n |(1-\gamma\lambda)^{n-k}(|\Delta_{k-1}|^2 - |\Delta_{k-1}^{(t)}|^2)|\} := \mathcal{T}_{n,t},$$

which can be further controlled by

$$\mathcal{T}_{n,t} \leq \gamma(1-\gamma\lambda)^{n-t}\mathbb{E}\{|Z_t|\sum_{k=1}^n |(1-\gamma\lambda)^{n-k}|(\Delta_{k-1} - \Delta_{k-1}^{(t)})(\Delta_{k-1} + \Delta_{k-1}^{(t)})|\}$$

$$\leq \gamma(1-\gamma\lambda)^{n-t}\sum_{k=1}^n (1-\gamma\lambda)^{n-k}\mathbb{E}\Big\{|Z_t||\theta_{k-1} - \theta_{k-1}^{(t)}|(|\theta_{k-1} - \theta^*| + |\theta_{k-1}^{(t)} - \theta^*|)\Big\}$$

We apply the Hölder inequality, for $k \geq t+1$,

$$\mathbb{E}\Big\{|Z_t||\theta_{k-1} - \theta_{k-1}^{(t)}|(|\theta_{k-1} - \theta^*| + |\theta_{k-1}^{(t)} - \theta^*|)\Big\}$$

$$\leq 2\|Z_t\|_4\|\Delta_{k-1}\|_4\psi(k-1,t)$$

$$\lesssim (\sqrt{\gamma} + (1-\alpha_0\gamma)^{(k-1)/2})\gamma(1-\alpha_0\gamma)^{(k-1-t)/2}$$

where the last inequality if from Theorem 2.3 and 3.1. For $k \leq t$, the expectation above is 0. Hence we have

$$\mathcal{T}_{n,t} \lesssim \gamma(1-\gamma\lambda)^{n-t}\sum_{k=t+1}^n (1-\gamma\lambda)^{n-k}(\sqrt{\gamma} + (1-\alpha_0\gamma)^{(k-1)/2})\gamma(1-\alpha_0\gamma)^{(k-1-t)/2},$$

$$= \gamma^2(1-\lambda\gamma)^{n-t}\sum_{k=t+1}^n \big[(1-\lambda\gamma)^{n-t/2-1} + \sqrt{\gamma}(1-\gamma\lambda)^{n-(k+t+1)/2}\big]$$

$$\leq \gamma^2(n-t)(1-\lambda\gamma)^{2n-3t/2-1} + \gamma^{5/2}\frac{(1-\lambda\gamma)^{(n-t-1)/2}}{1-\sqrt{1-\gamma\lambda}}$$

$$\lesssim \gamma^2(n-t)(1-\lambda\gamma)^{2n-3t/2-1} + \gamma^{3/2}(1-\lambda\gamma)^{(n-t-1)/2}.$$

Here we use the fact that $\sqrt{1-\lambda\gamma} \leq 1 - \lambda\gamma/2$. By elementary calculations,

$$\sum_{t=1}^n \mathbb{E}\{|\mathcal{I}_n - \mathcal{I}_n^{(t)}||\Gamma_n(\gamma)^{-1/2}\sqrt{\gamma}(\mathbf{I}_d - \gamma A)^{n-t}Z_t|\} \lesssim \sum_{t=1}^n \mathcal{T}_{n,t} \lesssim (1-\lambda\gamma)^{(n+1)/2} + \sqrt{\gamma}. \quad (30)$$

Similarly, by Cauchy inequality,

$$\mathbb{E}|\gamma^{-1/2}\Gamma_n(\gamma)^{-1/2}(I_{2,n} - I_{2,n}^{(t)})||\Gamma_n(\gamma)^{-1/2}\sqrt{\gamma}(\mathbf{I}_d - \gamma A)^{n-t}Z_t|$$

$$\lesssim (1-\gamma\lambda)^{n-t}\mathbb{E}\{|Z_t||I_{2,n} - I_{2,n}^{(t)}|\}$$

$$\leq (1-\gamma\lambda)^{n-t}\|Z_t\|_2\psi_D(n,t)$$

$$\lesssim \gamma(1-\lambda\gamma)^{(4n-3t-1)/2} + \gamma^{3/2}(1-\lambda\gamma)^{(3n-3t-1)/2}.$$

Summing the quantity above from $t = 1$ to $t = n$ yields an upper bound

$$\sum_{t=1}^n \mathbb{E}|\gamma^{-1/2}\Gamma_n(\gamma)^{-1/2}(I_{2,n} - I_{2,n}^{(t)})||\Gamma_n(\gamma)^{-1/2}\sqrt{\gamma}(\mathbf{I}_d - \gamma A)^{n-t}Z_t| \lesssim (1-\lambda\gamma)^{(n-1)/2} + \sqrt{\gamma}.$$

$$(31)$$

Then we consider the following error terms due to linear approximation:

$$\mathbb{E}\{|\Gamma_n(\gamma)^{-1/2}\sqrt{\gamma}(\mathbf{I}_d - \gamma A)^n\Delta_0||\Gamma_n(\gamma)^{-1/2}L_n/\sqrt{\gamma}|\},$$

$$\mathbb{E}\{|\Gamma_n(\gamma)^{-1/2}\sqrt{\gamma}I_{2,n}||\Gamma_n(\gamma)^{-1/2}L_n/\sqrt{\gamma}|\},$$

and

$$\mathbb{E}\{|\mathcal{I}_n||\Gamma_n(\gamma)^{-1/2}L_n/\sqrt{\gamma}|\}.$$

We use the Cauchy inequality to bound them. Notice that by definition of $\Gamma_n(\gamma)$, the sequence $\Gamma_n(\gamma)^{-1/2}L_n/\sqrt{\gamma}$ is standardized with fixed covariance matrix $\mathbf{I}_d$. For the first term,

$$\mathbb{E}\{|\Gamma_n(\gamma)^{-1/2}\sqrt{\gamma}(\mathbf{I}_d-\gamma A)^n\Delta_0||\Gamma_n(\gamma)^{-1/2}L_n/\sqrt{\gamma}|\}$$
$$\lesssim \sqrt{\gamma}(1-\lambda\gamma)^n\|\Delta_0\|_2\|\Gamma_n(\gamma)^{-1/2}L_n/\sqrt{\gamma}\|_2 \lesssim \sqrt{\gamma}(1-\lambda\gamma)^n. \tag{32}$$

For the second term,

$$\mathbb{E}\{|\Gamma_n(\gamma)^{-1/2}\sqrt{\gamma}I_{2,n}||\Gamma_n(\gamma)^{-1/2}L_n/\sqrt{\gamma}|\}$$
$$\lesssim \sqrt{\gamma}\|I_{2,n}\|_2\|\Gamma_n(\gamma)^{-1/2}L_n/\sqrt{\gamma}\|_2 \lesssim \gamma+\sqrt{\gamma}(1-\lambda\gamma)^{(n-1)/2}. \tag{33}$$

For the third term,

$$\mathbb{E}\{|\mathcal{I}_n||\Gamma_n(\gamma)^{-1/2}L_n/\sqrt{\gamma}|\}$$
$$\lesssim \sqrt{\gamma}\sum_{k=1}^n(1-\gamma\lambda)^{n-k}\mathbb{E}\{|\Delta_{k-1}|^2|\Gamma_n(\gamma)^{-1/2}L_n/\sqrt{\gamma}|\}$$
$$\leq \sqrt{\gamma}\sum_{k=1}^n(1-\gamma\lambda)^{n-k}\|\Delta_{k-1}\|_4^2$$
$$\lesssim \sqrt{\gamma}\sum_{k=1}^n(1-\gamma\lambda)^{n-k}[(1-\alpha_0\gamma)^{k-1}+\gamma]$$
$$\lesssim \sqrt{\gamma}+\sqrt{\gamma}n(1-\gamma\lambda)^{n-1}. \tag{34}$$

Combining all upper bounds of (29)-(34) and plugging them into the inequality (27) yields

$$\sup_{\mathcal{D}\in\mathcal{V}}|\mathbb{P}(\Gamma_n(\gamma)^{-1/2}\frac{\Delta_n}{\sqrt{\gamma}}\in\mathcal{D})-\mathbb{P}(\mathcal{N}(0,\mathbf{I}_d)\in\mathcal{D})|$$
$$\leq C(\sqrt{\gamma}+(1-\lambda\gamma)^{\frac{n-1}{2}}+\sqrt{\gamma}n(1-\lambda\gamma)^{n-1})$$

for some constant $C$ independent of $n$, $\gamma$ and $\theta_0$. By the discussion of Remark 1 in Samsonov et al. [2024], since $\Gamma_n(\gamma)^{-1/2}$ is non-degenerate, and an image of a convex set under a non-degenerate linear mapping is a convex set, we have

$$\sup_{\mathcal{D}\in\mathcal{V}}|\mathbb{P}(\Gamma_n(\gamma)^{-1/2}\frac{\Delta_n}{\sqrt{\gamma}}\in\mathcal{D})-\mathbb{P}(\mathcal{N}(0,\mathbf{I}_d)\in\mathcal{D})|=\sup_{\mathcal{D}\in\mathcal{V}}|\mathbb{P}(\frac{\Delta_n}{\sqrt{\gamma}}\in\mathcal{D})-\mathbb{P}(\mathcal{N}(0,\Gamma_n(\gamma))\in\mathcal{D})|.$$

To complete the proof, we use Theorem 1.1 in Devroye et al. [2018] (or Lemma 13 in Samsonov et al. [2024]), which bounds the total variation distance of two Gaussian measures by the distance between their covariance matrix. Here we only need to bound the convex distance, i.e.,

$$\sup_{\mathcal{D}\in\mathcal{V}}|\mathbb{P}(\mathcal{N}(0,\Gamma_n(\gamma))\in\mathcal{D})-\mathbb{P}(\mathcal{N}(0,\Gamma)\in\mathcal{D})|\leq\frac{3}{2}\|\Gamma^{-1/2}\Gamma_n(\gamma)\Gamma^{-1/2}-\mathbf{I}_d\|_F,$$

which trivially holds by Theorem 1.1 in Devroye et al. [2018]. Since matrix norms are equivalent, by (25) we have $\|\|\Gamma^{-1/2}\Gamma_n(\gamma)\Gamma^{-1/2}-\mathbf{I}_d\|_F \lesssim \gamma+(1-\gamma\lambda)^{2n}$. The last step is to use the triangular inequality of the convex distance, and the proof is completed. $\qquad\square$

## A.5 Proof of Nagaev-type Inequality

We first prove the results of non-linear auto-regressive approximation.

*Proof of Lemma 4.1.* Similar to the argument of the proof of Theorem 2.3,

$$\|\beta_n-\theta^*\|_p^2 \leq \|\beta_{n-1}-\theta^*-\gamma\nabla G(\beta_{n-1})\|_p^2+(p-1)\gamma^2\|Z_n\|_p^2$$
$$\leq (1-2\gamma\mu+(6p-5)\gamma^2L_p^2)\|\beta_{n-1}-\theta^*\|_p^2+3(p-1)\gamma^2\|Z_n\|_p^2. \tag{35}$$

Recursively updating it, we get

$$\|\beta_n - \theta^*\|_p^2 \le \frac{3(p-1)M_p^2}{\alpha_0}\gamma.$$

For the second result,

$$\|\theta_n - \beta_n\|_p^2$$

$$\le \|\theta_{n-1} - \beta_{n-1} - \gamma[\nabla G(\theta_{n-1}) - \nabla G(\beta_{n-1})]\|_p^2 \tag{36}$$

$$+ (p-1)\gamma^2\|\nabla G(\theta_{n-1}) - \nabla g(\theta_{n-1}, X_n) + Z_n\|_p^2$$

$$\le \left\|\langle\theta_{n-1} - \beta_{n-1}, \theta_{n-1} - \beta_{n-1}\rangle - 2\gamma\langle\theta_{n-1} - \beta_{n-1}, \nabla G(\theta_{n-1}) - \nabla G(\beta_{n-1})\rangle\right\|_{p/2}$$

$$+ \gamma^2\|\nabla G(\theta_{n-1}) - \nabla G(\beta_{n-1})\|_p^2 + 4(p-1)\gamma^2 L_p^2\|\theta_{n-1} - \theta^*\|_p^2 \tag{37}$$

$$\le (1 - 2\gamma\mu + \gamma^2 L_p^2)\|\theta_{n-1} - \beta_{n-1}\|_p^2 + 4(p-1)\gamma^2 L_p^2[(1 - \alpha_0\gamma)^n\|\theta_0 - \theta^*\|_p^2$$

$$+ 3(p-1)M_p^2\alpha_0^{-1}\gamma^3] \tag{38}$$

$$\le (1 - \alpha_1\gamma)\|\theta_{n-1} - \beta_{n-1}\|_p^2 + 4(p-1)\gamma^2 L_p^2[(1 - \alpha_0\gamma)^n\|\theta_0 - \theta^*\|_p^2$$

$$+ 3(p-1)M_p^2\alpha_0^{-1}\gamma^3], \tag{39}$$

$$\tag{40}$$

where $\alpha_1 \le 2\mu - \gamma L_p^2$ and $\alpha_1 \le \gamma^{-1}$. Without loss of generality, we can choose $\alpha_0 \le \alpha_1$ since the upper constraint of $\alpha_0$ is more stringent. Recursively updating the inequality, we get

$$\|\theta_n - \beta_n\|_p^2 \le [(1 - \alpha_1\gamma)^n + 4(p-1)L_p^2 n\gamma^2(1 - \alpha_0\gamma)^n]\|\Delta_0\|_p^2 + \frac{3(p-1)M_p^2}{\alpha_0\alpha_1}\gamma^2.$$

$\square$

Now we are ready to prove the sharp concentration inequality.

*Proof of Theorem 4.2.* Denote $\mathbb{E}_0 X = X - \mathbb{E}X$ as the centralized random variable $X$. Without loss of generality, we let $\alpha_0 \le \alpha_1$ in the following proof. Applying Markov inequality on Lemma 4.1, we have

$$\mathbb{P}(|\Delta_n - (\beta_n - \theta^*)| > \epsilon) \tag{41}$$

$$= \mathbb{P}(|\theta_n - \beta_n| > \epsilon) \tag{42}$$

$$\le \frac{\|\theta_n - \beta_n\|_p^p}{\epsilon^p} \tag{43}$$

$$\le \frac{2^{p-1}\rho_p^p(1 - \alpha_0\gamma)^{np/2}}{\epsilon^p}[1 + 4(p-1)L_p^2 n\gamma^2]^{p/2} + \frac{2^{p-1}\gamma^p(3p-3)^{p/2}M_p^p}{\epsilon^p(\alpha_0\alpha_1)^{p/2}}. \tag{44}$$

Then we consider the tail probability of $\mathbb{E}_0\{v^\top(\beta_n - \theta^*)\}$. Define $\mathcal{P}_k(\xi) = \mathbb{E}(\xi|\mathcal{F}_k) - \mathbb{E}(\xi|\mathcal{F}_{k-1})$ as the projection operator. Let $Z_i' = \nabla g(\theta^*, X_i')$ be the i.i.d. copy of $Z_i$. Similar to the proof of Theorem 3.1, we can show that for $1 \le k \le n$,

$$|\mathcal{P}_k(v^\top(\beta_n - \theta^*))| \le (1 - \alpha_0\gamma)^{n-k}\gamma\mathbb{E}(|Z_k - Z_k'||\mathcal{F}_k). \tag{45}$$

Let $y = p\epsilon/(p+2)$. Define the following sequence

$$\eta_i = \eta_{i-1} - \gamma\nabla G(\eta_{i-1}) - \gamma Z_i \times \min\left\{1, \frac{y}{2\gamma(1 - \alpha_0\gamma)^{n-i}|Z_i|}\right\}, \quad \eta_0 = \theta^*. \tag{46}$$

Then we have

$$\mathbb{P}(|\mathbb{E}_0\{v^\top(\beta_n - \theta^*)\}| > \epsilon) \tag{47}$$

$$\le \sum_{i=1}^n \mathbb{P}\left(|Z_i| > \frac{y}{2\gamma(1 - \alpha_0\gamma)^{n-i}}\right) + \mathbb{P}(|\mathbb{E}_0\{v^\top(\eta_n - \theta^*)\}| > \epsilon), \tag{48}$$

$$\le \frac{2^p M_p^p}{y^p}\gamma^p \sum_{i=1}^n (1 - \alpha_0\gamma)^{p(n-i)} + \mathbb{P}(|\mathbb{E}_0\{v^\top(\eta_n - \theta^*)\}| > \epsilon) \tag{49}$$

$$\le \frac{C\gamma^{p-1}}{\epsilon^p} + \mathbb{P}(|\mathbb{E}_0\{v^\top(\eta_n - \theta^*)\}| > \epsilon), \tag{50}$$

for some constant $C$ independent of $\epsilon$, $n$ and $\gamma$. The next step is to investigate the tail behavior of $\mathbb{E}_0\{v^\top(\eta_n - \theta^*)\}$. We consider its moment generating function: for $x > 0$,

$$\mathcal{M}_n(x) := \mathbb{E}\exp(x\mathbb{E}_0\{v^\top(\eta_n - \theta^*)\}) = \mathbb{E}\exp\left\{x\sum_{k=1}^n \mathcal{P}_k(v^\top(\eta_n - \theta^*)\right\}.$$

Due to definition (46) and a similar argument to (45), we have $\sup_{v\in\mathbb{S}^d}|\mathcal{P}_k(v^\top(\eta_n - \theta^*)| \leq y$. As a result, we can leverage Lemma 1.4 in [Nagaev, 1979] to obtain

$$\mathbb{E}\left\{\exp\left(x\mathcal{P}_k(v^\top(\eta_n - \theta^*))\right)\big|\mathcal{F}_{k-1}\right\}$$

$$\leq 1 + \frac{\exp(p)x^2}{2}\mathbb{E}\left\{|\mathcal{P}_k(v^\top(\eta_n - \theta^*))|^2\big|\mathcal{F}_{k-1}\right\}$$

$$+ \frac{\exp(xy) - 1 - xy}{y^p}\mathbb{E}\left\{|\mathcal{P}_k(v^\top(\eta_n - \theta^*))|^p\big|\mathcal{F}_{k-1}\right\} \times \mathbb{I}\left\{x > \frac{p}{y}\right\}$$

$$\leq 1 + \exp(p)x^2(1 - \alpha_0\gamma)^{2(n-k)}\gamma^2 M_2^2 + \frac{\exp(xy) - 1 - xy}{y^p}\frac{(1 - \alpha_0\gamma)^{p(n-k)}\gamma^p M_p^p}{2^p} \times \mathbb{I}\left\{x > \frac{p}{y}\right\}.$$

Since the final upper bound does not depend on $\mathcal{F}_{k-1}$, we have

$$\mathcal{M}_n(x) \leq \prod_{t=0}^{n-1}\left(1 + e^p x^2(1 - \alpha_0\gamma)^{2t}\gamma^2 M_2^2 + \frac{e^{xy} - 1 - xy}{y^p}2^p(1 - \alpha_0\gamma)^{pt}\gamma^p M_p^p \times \mathbb{I}\{x > p/y\}\right)$$

$$\leq \exp\left(C_p x^2 M_2^2 \gamma + C_p \gamma^{(p-1)}\frac{e^{xy} - 1 - xy}{y^p}M_p^p \times \mathbb{I}\{x > p/y\}\right),$$

for some constant $C_p$ independent of $\epsilon$, $n$ and $\gamma$. We use the Chernoff-type bound:

$$\mathbb{P}(|\mathbb{E}_0\{v^\top(\eta_n - \theta^*)\}| > \epsilon) \leq e^{-x\epsilon}\mathcal{M}_n(x),$$

and find an $x > 0$ such that

$$C_p x^2 M_2^2 \gamma - \frac{2y}{q} + C_p \gamma^{(p-1)}\frac{e^{xy} - 1 - xy}{y^p}M_p^p \times \mathbb{I}\{x > p/y\} - xy$$

is small. The calculation is identical to the proof of Theorem 1.3 in Nagaev [1979], which leads to

$$\mathbb{P}(|\mathbb{E}_0\{v^\top(\eta_n - \theta^*)\}| > \epsilon) \leq \frac{2}{C_p\epsilon^p M_p^p \gamma^{1-p} + 1} + 2\exp(-\frac{C_p\epsilon^2}{\gamma M_2^2}) \leq \frac{C_1\gamma^{p-1}}{\epsilon^p} + 2\exp(-\frac{C_2\epsilon^2}{\gamma})$$

$$\tag{51}$$

for some constant $C_1$ and $C_2$ independent of $\epsilon$, $n$ and $\gamma$.

Finally, we use union bound on (44), (50), and (51) to finish the proof of the tail probability inequality in Theorem 4.2. Notice that from Lemma 4.1 we have

$$|\mathbb{E}(\beta_n - \theta^*)| \leq \|\beta_n - \theta^*\|_p \leq M_p\sqrt{\frac{3(p-1)}{\alpha_0}}\sqrt{\gamma}.$$

As long as $M_p\sqrt{\frac{3(p-1)}{\alpha_0}}\sqrt{\gamma} < \epsilon$, this expectation term can be ignored in the tail probability bound. If not, i.e., $\epsilon^2/\gamma \leq 3M_p^2(p-1)/\alpha_0$, we can choose $C_2 \leq \log 2\alpha_0/3M_p^2(p-1)$ such that the probability bound trivially holds in this case. As a result, the expectation term does not affect the validity of our result, and the proof is completed. □

## A.6 Additional Experiment Details

We conducted the experiments in R version 4.3.1 (2023-06-16) on a MacBook Air with a GPU Apple M1, 4 performance and 4 efficiency cores, and 8 GB LPDDR4 memory, equipped with macOS Big Sur version 11.5.1.

