# OpenReview forum: "Gaussian Approximation and Concentration of Constant Learning-Rate Stochastic Gradient Descent"
_NeurIPS.cc/2025/Conference — NeurIPS 2025 poster_

### Official Review · Reviewer_oyBA · 2025-06-11

**Clarity:** 3
**Significance:** 3
**Originality:** 3
**Rating:** 4
**Confidence:** 4

**Summary:**

This paper presents a  theoretical analysis of stochastic gradient descent (SGD) with a constant learning rate. Unlike traditional analyses that require decaying step sizes or stationarity assumptions, the authors provide both finite-sample and asymptotic results under milder conditions. The key contributions include:

1. The authors develop a new linear approximation technique that enables a CLT for SGD iterates starting from arbitrary initialization, without requiring convergence to stationarity. The fluctuations of the iterates around the target parameter converge to a multivariate Gaussian distribution as the step size \\(\gamma \to 0\\).

 2.  The paper derives the  Berry–Esseen-type bound for constant-rate SGD, quantifying the distance between the SGD distribution and its Gaussian limit. The bound shows the approximation error is of order $\mathcal{O}(\sqrt{\gamma})$, which is optimal given the non-vanishing bias of the SGD iterates.

  3.  The authors provide explicit upper bounds for the $p$-th moments of the error $\mathbb{E}[|\theta_n - \theta^*|^p]$, establishing how higher-order moments decay with the number of iterations.

  4.  By approximating the SGD dynamics with a nonlinear autoregressive process, the paper derives sharp large-deviation bounds that combine sub-Gaussian and polynomial decay terms, allowing high-probability guarantees even in heavy-tailed noise settings.

  5.  Numerical experiments on logistic regression with heavy-tailed data validate the theoretical bounds. The proposed Nagaev-type inequality closely matches the empirical tail probabilities and significantly outperforms Markov-type bounds in tightness.

**Questions:**

1. How do the assumptions in this paper (e.g., strong convexity, Lipschitz continuity) limit the applicability of the results to non-convex optimization problems commonly encountered in deep learning?

2.  The paper presents a quenched central limit theorem that applies to non-stationary SGD. How does this differ from classical CLTs for SGD that require asymptotic stationarity? What practical advantage does this offer?

3.  Why is the Berry--Esseen bound particularly important for understanding the behavior of SGD in finite-sample regimes? How does this bound compare to previous results for decaying learning rates?  The matrix $\Gamma$ is typically unknown and should be estimated.  How are the authors going to estimate it.

4.  The simulations focus on logistic regression with heavy-tailed inputs. Would the theoretical bounds hold or degrade for more complex, non-linear models (e.g., neural networks)?

 5.  The authors claim their bounds are sharp. What empirical or theoretical evidence supports this, and under what conditions might the bounds be loose or overly conservative?

 6.  The paper discusses optimal scaling of $\gamma$ in terms of $n$ for convergence. How should practitioners choose $\gamma$ in practice based on this theory, especially when $n$ is unknown?

**Ethical Concerns:**

["NO or VERY MINOR ethics concerns only"]

**Final Justification:**

Thank you for the detailed clarifications. Although I remain unconvinced of the immediate practicality of the obtained results, the theoretical foundations laid here are important. Accordingly, I will maintain my high rating.

**Limitations:**

Yes

**Paper Formatting Concerns:**

No issues

**Quality:**

3

**Strengths And Weaknesses:**

Strengths
1.  The paper establishes the first Berry--Esseen bound and Nagaev-type inequality for constant learning-rate SGD.
 2. The central limit theorem applies to non-stationary SGD sequences with arbitrary initialization, which is more aligned with practical scenarios.
 3. The work provides moment bounds and concentration inequalities that are valid for finite samples.
 4. The use of linear and autoregressive approximations, functional dependence measures, and Lyapunov equations is mathematically rigorous and innovative.
 5.  Numerical studies effectively support the theoretical claims, demonstrating the sharpness of the proposed tail bounds.



Weaknesses

1. The simulations are confined to a single logistic regression setting, limiting insight into performance across diverse models or datasets.

 2.  Although milder than prior work, the assumptions (e.g., strong convexity, Lipschitz continuity) may still be restrictive for many non-convex or real-world scenarios.

 3. The focus is on theoretical analysis; no new algorithm or implementation technique is proposed, which may limit the paper’s immediate practical impact.

 4.  There is no discussion on computational efficiency or how the proposed bounds scale with dimensionality in practical settings.

5. It is not clear how the authors estimate the covariance matrix $\Gamma$ and how this would influence the final bounds.

---

> ### Author Rebuttal · Authors · 2025-07-31
>
> Thank you very much for your thoughtful comments and for raising these insightful questions. Please see our point-to-point responses to your concerns below.
>
>
>
> $\textbf{Weakness}$
>
>  $\textbf{W1:}$ We recognize the experiment section as you commented, since our paper is more of a theoretical nature. We will add different model scenarios in our updated manuscript.
>
>
>
>  $\textbf{W2:}$ Without convexity, the convergence behavior of SGD remains largely unexplored. Generally, we do not think the asymptotic distribution will be Gaussian. The iterates do not necessarily converge to a unique global optimum—let alone admit a well-defined asymptotic distribution or central limit theorem. However, under assumptions milder than global convexity, it is possible to extend the approach in this paper to obtain similar results. For example, if the objective has multiple local minimizers and is locally strongly convex around each minimum, we expect the asymptotic distribution of SGD to be a mixture of Gaussian distributions with centers at local minima.
>
>  $\textbf{W3:}$ We agree that due to the theoretical nature of our paper, we did not propose new algorithm. However, we still discuss some implementation technique, for instance, the choice of learning rate $\gamma = \mathcal{O}(\log n /n)$ suggested by the Gaussian approximation result. Our work sheds new light on important practical concerns, such as the validity of finite-sample statistical inference and sample complexity.
>
>    $\textbf{W4:}$  We would like to clarify that, since we did not propose new algorithm, the only algorithm involved in our paper is the classical SGD, for which the computational efficiency is well-studied and does not belong to the scope of this work.
>
> We fully acknowledge that how the bounds scale with dimensionality is a pertinent point. This paper mainly focuses on the convergence rate w.r.t. learning rate $\gamma$ and the sample size $n$. In fact, we have discussed the dependence of the Nagaev inequality after line 255. For the Berry-Esseen bound, based on other results and techniques related to the Gaussian approximation of SGD (e.g., Sheshukova et al. (2025), Theorem 2), we expect our Berry-Esseen bound to scale as $\sqrt{d}$. It is an interesting future work to optimize the dimension dependence result for SGD, as the best order of dimension for the independent sum is $d^{1/4}$ (Raivc 2019).
>
>  $\textbf{W5:}$ Thank you for raising this thoughtful question. The covariance and precision matrix estimation for stationary processes or SGD sequences has been extensively researched. For instance, if the Hessian information $\nabla^2 g(\theta_{n-1},X_n)$, we can use Plug-in estimator in [R1] to separately estimate $A$ and $S$ by $\hat{A}$ and $\hat{S}$, and numerically solve $\Gamma\hat{A}+ \hat{A}\Gamma=\hat{S}$. Please refer to Section 4.1 in [R1] for detailed implementation. Another choice is to estimate $\Gamma$ using only SGD iterates. Since constant learning rate SGD converges to its stationary distribution exponentially fast (Dieuleveut et al. 2020), we can leverage those developed estimation approaches for stationary processes, e.g., the thresholded estimator and graphical Lasso estimator in [R2]. These approaches can directly estimate $\Gamma$ and its inverse.
>
> When comparing the distribution of
> $\Delta_{n}/\sqrt{\gamma}$
> with that of the multivariate Gaussian random vector
> $N(0, \hat{\Gamma})$, a key step is to establish a Gaussian comparison result that provides a sharp non-asymptotic upper bound for the Kolmogorov distance
>  $\sup_{\mathcal{D} \in \mathcal{V}} |\mathbb{P}(N(0, \Gamma) \in \mathcal{D}) - \mathbb{P} (N(0, \hat{\Gamma}) \in \mathcal{D})|$.
>
> Such Gaussian comparison bounds have been recently derived in specific settings: for example, when $\mathcal{V}$ is the class of rectangles in $\mathbb{R}^{d}$ (Chernozhukov et al. (2015)), and when $\mathcal{V}$ is the class of closed balls in $\mathbb{R}^{d}$ (Gotze et al. (2019)). We believe that obtaining sharp upper bounds for $\sup_{\mathcal{D} \in \mathcal{V}} |\mathbb{P}(N(0, \Gamma) \in \mathcal{D}) - \mathbb{P} (N(0, \hat{\Gamma}) \in \mathcal{D})|$ is both an interesting and important problem in probability theory, and also serves as a valuable tool for uncertainty quantification in stochastic algorithms. However, this question lies beyond the scope of the present work, and we would leave it for future work.
>
> $\textbf{References}$
>
> [1] Chen, X., J. D. Lee, X. T. Tong, and Y. Zhang (2020). “Statistical inference for model parameters in
> stochastic gradient descent”. In: The Annals of Statistics 48.1, pp. 251–273
>
> [2] Chen, X., M. Xu, and W. B. Wu (2013). “Covariance and precision matrix estimation for high-dimensional
> time series”. In: The Annals of Statistics 41.6, pp. 2994–3021. doi: 10.1214/13-AOS1182.
>
>
>
>
>
> $\textbf{Questions}$
>
> $\textbf{Q1:}$  Without convexity, the convergence behavior of SGD remains largely unexplored. Generally, we do not think the asymptotic distribution will be Gaussian. The iterates do not necessarily converge to a unique global optimum—let alone admit a well-defined asymptotic distribution or central limit theorem. However, under assumptions milder than global convexity, it is possible to extend the approach in this paper to obtain similar results. For example, if the objective has multiple local minimizers and is locally strongly convex around each minimum, we expect the asymptotic distribution of SGD to be a mixture of Gaussian distributions with centers at local minima.
>
>
> $\textbf{Q2:}$ Thank you for the insightful questions. In one word, the classical CLT requires the starting point $\theta_0$ to follow the stationary distribution, while the quenched central limit theorem is applicable to arbitrary initialization. In analyzing stochastic approximation algorithms, we find that the quenched central limit theorem (CLT), which characterizes the conditional distribution given any initialization $\theta_{0}$, is more suitable than classical (annealed) CLTs that rely on the asymptotic stationarity. In particular, by conditioning on the initialization, the classical CLT becomes a direct corollary of the quenched CLT, which establishes the asymptotic normality of SGD iterates for arbitrary initializations. Practically, this allows greater flexibility in choosing initial estimates, as the quenched CLT guarantees asymptotic normality without imposing any restriction on initialization.
>
>
>
> $\textbf{Q3:}$ Thank you for the insightful questions. The key advantage of the Berry-Esseen bound lies in its non-asymptotic nature: it provides a quantitative bound on the Kolmogorov distance between the distribution of SGD iterates and the normal distribution. This is particularly valuable in finite-sample regimes, similar to sample complexity analysis, on how many training samples are needed to achieve a desired level of distributional approximation accuracy.
>
> As we discussed from lines 220 to 223, the bound for the constant learning rate is sharper than the result for the decaying learning rate. By choosing $\gamma = \mathcal{O}(\log n/n)$, the order of Gaussian approximation becomes $\mathcal{O}(\sqrt{\log n/n})$, while the optimal approximation rate for decaying step size is $n^{-1/4}$. For the covariance estimation, please kindly refer to our reply above (to the weakness 5).
>
>
>
>  $\textbf{Q4:}$The theoretical bounds hold for any models satisfying Assumptions 2.1 and 2.2, including more complex, non-linear models. In general, neural networks do not satisfy the convexity, so our result is not applicable. However, as we mentioned before, it is possible to extend the approach in this paper to obtain similar results for non-convex loss functions.
>
> $\textbf{Q5:}$ Thank you for your thoughtful comments and for raising this insightful question. To illustrate the sharpness of our Berry-Esseen bound in Theorem 3.4, let $X_{1}, X_{2}, \ldots$ be i.i.d.~centered random variables with $\mathbb{E} (X_{1}^{2}) = 1$ and $\mathbb{E} |X_{1}|^{3} < \infty$. We consider estimating
>
> $\theta^{*} = \mathbb{E} (X\_{1})$
>
> using the SGD algorithm with
> $\theta_{0} = \theta^{*}$
> and
> $\theta_{i} = \theta_{i - 1} - \gamma (\theta_{i - 1} - X_{i})$
> for $i \geq 1$.
>
> Then $\theta_{n}$ admits the linear expansion $\Delta_{n} = \gamma \sum_{i = 1}^{n} (1 - \gamma)^{n - i} (X_{i} - \theta^{*})$ which is a weighted sum of independent centered random variables. Consequently, for any $z \in \mathbb{R}$, the classical Edgeworth expansion yields
>     \begin{equation}
>         \mathbb{P} (\frac{\Delta_{n}}{\sqrt{\gamma/2}} \leq z) = \Phi(z) + \frac{\sqrt{2}\mathbb{E} (X_{1}^{3}) (z^{2} - 1) \phi(z)}{9} \sqrt{\gamma} + o(\sqrt{\gamma}),
>     \end{equation}
>     where $\phi(\cdot)$ and $\Phi(\cdot)$ denote the probability density function and cumulative distribution function of the standard normal distribution, respectively. This expansion demonstrates that the leading $\sqrt{\gamma}$ term in our Berry-Esseen bound is sharp and unimprovable in general.
>
>
> $\textbf{Q5:}$Thanks a lot for this great question. We strongly agree that $n$ is usually unknown. In practice, a small choice of $\gamma$ ranging from $10^{-2}$ to $10^{-5}$ is very popular (see for example, Krizhevsky et al. (2012); Iiduka (2022) and the tutorial of PyTorch SGD ``torch.optim''), but the theoretical asymptotic behavior of constant learning-rate SGD as $\gamma$ close to zero is rarely studied in the literature. Our work fills in this gap by providing rigorous (non)-asymptotic theory for the SGD iterates. For such small $\gamma$, we expect the normal approximation works.

---

> > ### Comment · Reviewer_oyBA · 2025-08-04
> >
> > Thank you for the detailed clarifications. Although I remain unconvinced of the immediate practicality of the obtained results, the theoretical foundations laid here are  important. Accordingly, I will maintain my rating.

---

> ### Author Response · Authors · 2025-08-09
>
> We sincerely appreciate your thorough review. Thank you again for your recognition of our clarifications and contributions!

---

### Official Review · Reviewer_ZNkP · 2025-06-15

**Clarity:** 4
**Significance:** 4
**Originality:** 3
**Rating:** 6
**Confidence:** 4

**Summary:**

The paper provides a Berry-Esseen bound for the last-iterate error of SGD, with an $\tilde{\mathcal O}(n^{-1/2})$ (dimension dependence suppressed) normal approximation error over all convex sets in $\mathbb R^d$.

**Questions:**

- Can the authors spell out the dependence of $C$ on other problem parameters, in particular the dimension $d$, after Theorem 3.4?

- Related, it is known in the i.i.d.\ random vector case that by considering all closed balls instead of all convex sets in $\mathbb R^d$, the Berry-Esseen bound's $O(d^{1/4})$ dependence on $d$ can be removed. Can the authors achieve the same via taking $\mathcal V$ to be all closed balls?

- Would the authors be able to say a few things about the BE analysis of the Polyak-Ruppert averaged iterate? The PR average has a smaller variance in its limiting Gaussian distribution, right? So its BE approx might be of more practical interest.

**Ethical Concerns:**

["NO or VERY MINOR ethics concerns only"]

**Final Justification:**

We thank the authors for addressing these minor questions that I asked. I would like to keep my strong endorsement of this work unchanged.

**Limitations:**

Yes

**Paper Formatting Concerns:**

I have a minor concern that the primary area of this paper should be optimization instead.

**Quality:**

4

**Strengths And Weaknesses:**

The paper is very crisply written, well structured, and bears a very elegant and important result. While it is long known that the last iterate of SGD is asymptotically normal, this paper's quantitative version of such asymptotically normality opens up numerous downstream inference/uncertain quantification tasks.

I only have good things to say about this paper. A minor weakness might be that some potentially interesting dependences, in particular the one on $d$, are suppressed while directly porting the BE result by Shao & Zhang.

---

> ### Author Rebuttal · Authors · 2025-07-31
>
> We are extremely grateful for your strong endorsement and careful review of our proofs. Please see below our response to your questions.
>
> **1. I only have good things to say about this paper. A minor weakness might be that some potentially interesting dependence, in particular the one on $d$, are suppressed while directly porting the BE result by Shao \& Zhang. Can the authors spell out the dependence of $C$ on other problem parameters, in particular the dimension $d$, after Theorem 3.4?**
>
> **Answer:** We fully acknowledge that an explicit form of the constant depending on $d$ and other parameters will enhance the current result. Based on other results and techniques related to the Gaussian approximation of SGD (e.g., Sheshukov et al. 2025, Theorem 2), we expect our Berry-Esseen bound to scale as $\sqrt{d}$. It is an interesting open problem whether BE for SGD can achieve or approximate the order $d^{1/4}$. The expression of $C$ by other problem parameters requires more elementary calculation. We can derive it and give an explicit form in our updated manuscript.
>
> **2. Related, it is known in the i.i.d.\ random vector case that by considering all closed balls instead of all convex sets in $\mathbb{R}^d$, the Berry-Esseen bound's $O(d^{1/4})$ dependence on $d$ can be removed. Can the authors achieve the same via taking $\mathcal{V}$ to be all closed balls?**
>
> **Answer: Thank you for your valuable comments and for raising this interesting question. Yes, the techniques we developed for deriving the Berry-Esseen bound over all convex sets can indeed be extended to the case of closed balls. Specifically, by combining our refined linear approximation result from Section 3.2 with either the classical Berry-Esseen theorem (Bentkus and Gotze (1996)) or modern $L^{2}$-type Gaussian approximation result (Xu, Zhang and Wu (2019)), we can derive Berry-Esseen bounds over all closed balls that do not involve the dimensionality $d$. We are currently actively working on deriving the explicit form of this Berry-Esseen upper bound. **
>
> **3. Would the authors be able to say a few things about the BE analysis of the Polyak-Ruppert averaged iterate? The PR average has a smaller variance in its limiting Gaussian distribution, right? So its BE approx might be of more practical interest.**
>
> **Answer:** Thank you for the insightful question. We agree that, in general, the variance of Polyak-Ruppert averaged SGD is smaller than that of the vanilla SGD iterate. We expect the Berry-Esseen bound will be sharper. The theoretical BE analysis of the Polyak-Ruppert averaged iterate differs from our paper only in some elementary calculations. Current arguments of proof will follow similarly with modest adjustments.
>
> **4. I have a minor concern that the primary area of this paper should be optimization instead.**
>
> **Answer:** Thank you for your observation. We appreciate your perspective and are open to reconsidering the primary area of the paper.

---

### Official Review · Reviewer_ar9s · 2025-06-30

**Clarity:** 4
**Significance:** 4
**Originality:** 3
**Rating:** 4
**Confidence:** 3

**Summary:**

This work studies the distribution of SGD iterates under a constant learning rate. They prove that under certain assumptions such as strong convexity and Lipschitz, fluctuations around its mean converge to a multivariate normal distribution. Then they find the error of convergence through Barry-Essen bound in Gaussian approximation, characterize the large tail deviation using Nagaev type inequalities and find condition for convergence of higher order moments. Deviating from past works, that require a stationary distribution, this works provides the results for a finite step-size reflecting practical behaviour.

**Questions:**

Please see points 2 and 3.

**Ethical Concerns:**

["NO or VERY MINOR ethics concerns only"]

**Final Justification:**

I have read the rebuttal. Based on the merits, such as theoretical strength and empirical weakness, I stick to my score.

**Limitations:**

As I said, the assumptions on the loss function are too restrictive. The most basic loss to analyze the convergence is a quadratic, which the authors skipped to for a logistic regression loss. I would suggest l2 regularized logistic regression since it is strongly convex, as a better alternate experiment.

**Paper Formatting Concerns:**

None.

**Quality:**

4

**Strengths And Weaknesses:**

This is an interesting paper that characterizes the converges of SGD iterates under certain conditions. Based on the research interest on practical deep learning such as finite-step size, I think this is a timely topic.

Strengths:

1) The paper is generally well written and the notations are explained carefully for ease of the reader.
2) The remarks after every theroem explains the role of each hyperparameter.

Weakness:

1) *Lack of convincing experiments*: It would have made the paper stronger if the distribution was analyzed for more loss functions. It is well known that logistic regression is not strongly convex, although the theoritical assumption has strong convexity in it. I think the basic loss function is a quadratic where both the strong convexity parameter and the lipschitz parameter depend on the eigenvalues of the Hessian, which is easily calculated. If one can characterize the distribution is on a simple quadratic, then that would validate all the theorems. Currently this is missing.

2) *What if stochastic noise is Gaussian*: If the noise in the gradient assumes a normal distribution and loss is quadratic, then the distribution of the SGD iterates I think should also be an exact normal distribution with 0 error in Barry Essen type inequality. Does the current theorem cover this case? For example in 3.4, will these two assumptions make the Barry-Essen bound go to zero?

3) *Question on loss surface assumptions*: Although strong convexity and lipschitz assupmtions are slightly strong, yet necessary for a nice result, I assume what happens for general ovreparameterized deep neural networks where these conditions do not hold (consider a saddle near a minima). What will the distribution of SGD iterates look like? Will they still converge to a Gaussian distribution? This is not necessarily a weakness but I want to hear the author's comments.

Regarding the originality, I am not aware of all the papers on characterizing the distribution convergence of SGD iterates, but I agree that a finite step-size is indeed an interesting result.

---

> ### Author Rebuttal · Authors · 2025-07-31
>
> Thank you very much for your thoughtful comments and for raising these insightful questions. Please see our point-to-point responses to your concerns below.
>
> **1. Lack of convincing experiments: It would have made the paper stronger if the distribution was analyzed for more loss functions. It is well known that logistic regression is not strongly convex, although the theoretical assumption has strong convexity in it. I think the basic loss function is a quadratic where both the strong convexity parameter and the Lipschitz parameter depend on the eigenvalues of the Hessian, which is easily calculated. If one can characterize the distribution is on a simple quadratic, then that would validate all the theorems. Currently this is missing.**
>
> **Answer:**  We recognize the experiment section as you commented, since our paper is more of a theoretical nature. We totally agree that some experiments on quadratic loss functions will be a good ingredient, and will expand this part in our updated manuscript.
>
> **2. What if stochastic noise is Gaussian: If the noise in the gradient assumes a normal distribution and loss is quadratic, then the distribution of the SGD iterates I think should also be an exact normal distribution with 0 error in Barry Essen type inequality. Does the current theorem cover this case? For example in 3.4, will these two assumptions make the Barry-Essen bound go to zero?**
>
> **Answer:** You are absolutely right: when the stochastic noise is Gaussian and the objective function is quadratic, the Kolmogorov distance is exactly zero. Our Theorem 3.4 does cover this scenario; however, it provides a universal upper bound rather than the exact (zero) error. This is because Berry-Esseen-type bounds are designed to be general-purpose, worst-case estimates that apply broadly to a wide class of distributions. Therefore, while the bound in Theorem 3.4 remains valid, it may not be tight in this special Gaussian-quadratic case.
>
> **3. Question on loss surface assumptions: Although strong convexity and Lipschitz assumptions are slightly strong, yet necessary for a nice result, I assume what happens for general ovreparameterized deep neural networks where these conditions do not hold (consider a saddle near a minima). What will the distribution of SGD iterates look like? Will they still converge to a Gaussian distribution? This is not necessarily a weakness but I want to hear the author's comments.**
>
> **Answer:** Without convexity, the convergence behavior of SGD remains largely unexplored. Generally, we do not think the asymptotic distribution will be Gaussian. The iterates do not necessarily converge to a unique global optimum—let alone admit a well-defined asymptotic distribution or central limit theorem. However, under assumptions milder than global convexity, it is possible to extend the approach in this paper to obtain similar results. For example, if the objective has multiple local minimizers and is locally strongly convex around each minimum, we expect the asymptotic distribution of SGD to be a mixture of Gaussian distributions with centers at local minima. The case including a saddle is an even more challenging problem and is beyond the scope of this paper. We believe CLT does not hold in this setting, and will leave this as an interesting follow-up problem.
>
> **4. Regarding the originality, I am not aware of all the papers on characterizing the distribution convergence of SGD iterates, but I agree that a finite step-size is indeed an interesting result.**
>
> **Answer:** Thank you for your approval! This is the first work to establish non-asymptotic Gaussian approximation and sharp Nagaev-type inequalities for the constant learning rate SGD.
>
> **5. As I said, the assumptions on the loss function are too restrictive. The most basic loss to analyze the convergence is a quadratic, which the authors skipped to for a logistic regression loss. I would suggest l2 regularized logistic regression since it is strongly convex, as a better alternate experiment.**
>
> **Answer:**  Thank you for your constructive review! We fully acknowledge your suggestion and will expand the experiment section following it.

---

> > ### Comment · Reviewer_ar9s · 2025-08-05
> > **Acknowledgement**
> >
> > I have read the rebuttal. Based on the merits, such as theoretical strength and empirical weakness, I stick to my score.

---

> > > ### Author Response · Authors · 2025-08-09
> > >
> > > We sincerely appreciate your careful review and your valuable feedback on the originality and significance of our paper. Thank you again for your support!

---

### Official Review · Reviewer_5tvP · 2025-07-02

**Clarity:** 2
**Significance:** 3
**Originality:** 3
**Rating:** 4
**Confidence:** 4

**Summary:**

This paper develops a detailed finite-sample and asymptotic theory for constant learning-rate SGD. By leveraging linear approximations and functional dependence techniques, it derives a quenched central limit theorem, a Berry-Esseen bound, and Nagaev type tail inequalities.

**Questions:**

1. What is the implication of the bound in Theorem 3.1? Furthermore, how does Theorem 3.1 contribute to the main results?

2. What is the definition of $\beta^*$  on page 7?

3. Expanding the current experiments to include more diverse scenarios, additional step sizes, and higher dimensional cases would strengthen the practical validation of the tail bound.

4. The Berry-Esseen bound in Theorem 3.4 is valuable and novel. However, its practical usefulness remains somewhat unclear. The paper does not discuss how the bound scales with dimension, nor does it provide numerical experiments to illustrate how tight or meaningful the bound is in practice. Including explicit dimension dependence, practical guidance on estimating the constants involved, and empirical validation would significantly strengthen the impact and applicability of this result.

5. The theoretical contributions of this work are valuable and novel, but the lack of explanatory context and practical examples significantly reduces readability and impact. With improvements in exposition and additional experiments, the paper could be a strong contribution.

**Ethical Concerns:**

["NO or VERY MINOR ethics concerns only"]

**Final Justification:**

The paper is solid with strong, novel theoretical contributions. However, the lack of explanatory context and practical examples limits its clarity and impact, and the experimental validation remains narrow in scope. Based on these considerations, I will maintain my current score.

**Limitations:**

yes

**Quality:**

3

**Strengths And Weaknesses:**

**Strengths:** This work introduces novel approximation techniques and establishes sharp finite sample guarantees, providing the first Berry–Esseen bound and non-asymptotic tail bounds for constant step size SGD. It advances our understanding of the non-stationary behavior of SGD beyond the classical results.


**Weaknesses:** The paper primarily lists various techniques and results, but it leaves much of the intuition and practical usefulness of these tools insufficiently explained. Moreover, the experimental section is quite limited, focusing only on verifying the Nagaev type tail inequality through synthetic logistic regression experiments in low dimensions, and testing just three learning rates. It does not explore higher-dimensional scenarios or examine other theoretical results, such as the Berry-Esseen bound or the central limit theorem.

---

> ### Author Rebuttal · Authors · 2025-07-31
>
> Thank you very much for your thoughtful comments and for raising these insightful questions! We truly appreciate your careful review. Below, we have addressed all your questions and will revise the paper accordingly. We warmly welcome any additional suggestions to further enhance our work. If our rebuttal has satisfactorily addressed your concerns, we would be grateful if you would consider increasing your score. Please see our point-to-point responses to your concerns below.
>
> **1. The paper primarily lists various techniques and results, but it leaves much of the intuition and practical usefulness of these tools insufficiently explained. Moreover, the experimental section is quite limited, focusing only on verifying the Nagaev type tail inequality through synthetic logistic regression experiments in low dimensions, and testing just three learning rates. It does not explore higher-dimensional scenarios or examine other theoretical results, such as the Berry-Esseen bound or the central limit theorem.**
>
> **Answer:** Thank you for your constructive comments! We indeed conduct simulations to validate the central limit theorem and Gaussian approximation results. In particular, we conduct multiple independent trials of SGD on the linear, logistic, and expectile regression models and varying dimensions. In each scenario, the vanilla SGD distributes close to Gaussian, which empirically validates the CLT of constant learning rate SGD. We will include this part in our updated manuscript.
>
> **2. What is the implication of the bound in Theorem 3.1? Furthermore, how does Theorem 3.1 contribute to the main results?**
>
> **Answer:** In the context of SGD, the functional dependence measure quantifies the contribution of the $t$-th training sample $X\_t$ to the $n$-th SGD iterate $\theta\_n$. In other words, it measures the degree of dependence of outputs $\theta\_n$ on inputs $X\_t$. Theorem 3.1 indicates that this degree of dependence decays exponentially fast for the constant learning rate SGD. The derivation of the Berry-Esseen inequality for SGD relies on the idea of linear approximation and coupling. Theorem 3.1 plays a vital role in the calculation of the dependence measure of the error terms from linear approximation, which constitute the final bound.
>
> **3. What is the definition of $\beta^*$ on page 7?**
>
> **Answer:** Thank you for pointing this out. This is a typo. Here $\beta^{*}$ should be $\theta^{*}$.
>
> **4. Expanding the current experiments to include more diverse scenarios, additional step sizes, and higher dimensional cases would strengthen the practical validation of the tail bound.**
>
> **Answer:** Thank you for the suggestion, and we will expand the experiments section in our updated version, as we replied before.
>
> **5. The Berry-Esseen bound in Theorem 3.4 is valuable and novel. However, its practical usefulness remains somewhat unclear. The paper does not discuss how the bound scales with dimension, nor does it provide numerical experiments to illustrate how tight or meaningful the bound is in practice. Including explicit dimension dependence, practical guidance on estimating the constants involved, and empirical validation would significantly strengthen the impact and applicability of this result.**
>
> **Answer:** We fully acknowledge that an explicit dimension dependence result will enhance the current result. This paper mainly focuses on the convergence rate w.r.t. learning rate $\gamma$ and the sample size $n$. Based on other results and techniques related to the Gaussian approximation of SGD (e.g., Sheshukov et al. 2025, Theorem 2), we expect our Berry-Esseen bound to scale as $\sqrt{d}$.  It is an interesting future work to optimize the dimension dependence result for SGD, as the best order of dimension for independent sum in $d^{1/4}$ (Raic 2019).
>
> To illustrate the sharpness of our Berry-Esseen bound in Theorem 3.4, let $X\_{1}, X\_{2}, \ldots$ be i.i.d. centered random variables with $\mathbb{E} (X\_{1}^{2}) = 1$ and $\mathbb{E} |X\_{1}|^{3} < \infty$. We consider estimating $\theta^{\*} = \mathbb{E} (X\_{1})$ using the SGD algorithm with $\theta\_{0} = \theta^{\*}$ and $\theta\_{i} = \theta\_{i - 1} - \gamma (\theta\_{i - 1} - X\_{i})$ for $i \geq 1$. Then $\theta\_{n}$ admits the linear expansion $\Delta\_{n} = \gamma \sum_{i = 1}^{n} (1 - \gamma)^{n - i} (X\_{i} - \theta^{\*})$ which is a weighted sum of independent centered random variables. Consequently, for any $z \in \mathbb{R}$, the classical Edgeworth expansion yields
>     \begin{equation}
>         \mathbb{P} (\frac{\Delta\_{n}}{\sqrt{\gamma/2}} \leq z) = \Phi(z) + \frac{\sqrt{2}\mathbb{E} (X\_{1}^{3}) (z^{2} - 1) \phi(z)}{9} \sqrt{\gamma} + o(\sqrt{\gamma}),
>     \end{equation}
> where $\phi(\cdot)$ and $\Phi(\cdot)$ denote the probability density function and cumulative distribution function of the standard normal distribution, respectively. This expansion demonstrates that the leading $\sqrt{\gamma}$ term in our Berry-Esseen bound is tight and unimprovable in general. We will conduct experiments to provide more empirical evidence on this.
>
> Practical concerns of the Berry-Esseen inequality are discussed in the Remark after Theorem 3.4. The Gaussian approximation rate suggests a choice of learning rate $\gamma = \mathcal{O}(\log n /n)$. It also enables nonasymptotic statistical inference, i.e., the validity of inference procedure with finite sample size and fixed learning rate. We agree that some guidance on estimating related parameters is a pertinent complement.
>
> **6. The theoretical contributions of this work are valuable and novel, but the lack of explanatory context and practical examples significantly reduces readability and impact. With improvements in exposition and additional experiments, the paper could be a strong contribution.**
>
> **Answer:** Thank you for your approval and valuable comments. We recognize it as being a bit dense, since it is more of a theoretical nature. However, we strive to make it more readable, and have included more explanations and remarks after your comment. We would also like to explain some high-level ideas behind the proof of the Nagaev inequality, Theorem 4.2. Nagaev (1979) establishes a sharp concentration inequality of independent sums relying on the multiplicativity of moment generating functions (MGF). However, the non-linearity and dependence structure of SGD becomes a roadblock, where the MGF can not be explicitly computed. To address this issue, we leverage a nonlinear autoregressive coupling idea, i.e., approximate the SGD iterates with a gradient descent iteration with i.i.d. noises (Lemma 4.1). The MGF of the coupling sequence is tractable, based on which we apply a Chernoff-type bound to obtain the result.

---

> > ### Comment · Reviewer_5tvP · 2025-08-03
> >
> > Thanks to the authors for their responses. I have no further questions for now.

---

> > > ### Author Response · Authors · 2025-08-09
> > >
> > > Thank you again for recognizing the novelty of our work and your valuable feedback! We are delighted that all your questions have been addressed.

---

### Official Review · Reviewer_AvJc · 2025-07-02

**Clarity:** 3
**Significance:** 4
**Originality:** 4
**Rating:** 6
**Confidence:** 3

**Summary:**

This paper derives a variety of quantitative statistical properties of SGD iterates when the averaged loss is strongly convex and the step size is constant. In particular, it gives variations on the $p$-th moment bounds, the central limit theorem, Berry-Esseen bounds, and large-deviation inequalities. The Berry-Esseen bounds and large deviation inequalities give quantitative bounds on the deviation of the iterates from Gaussian random variables.

**Questions:**

- Can the constants on the theorems be quantified?
- Would it be possible to modify the methods to handle non-strongly convex problems?
- In the simulation, values of constants associated with the asymptotic distribution are used, and this is also remarked earlier to suggest that part of the Nagaev-type bounds are tight. Could the use of the constants from the asymptotic distributions be justified rigorously?

**Ethical Concerns:**

["NO or VERY MINOR ethics concerns only"]

**Final Justification:**

The paper was good, and I already had given the highest possible rating. I am keeping my score after the rebuttal.

**Limitations:**

Yes.

**Quality:**

4

**Strengths And Weaknesses:**

# Strengths
- The results give a more more detailed statistical analysis of SGD than currently available. While basic convergence properties are well-understood, the results in this paper give precise quantitative bounds on the deviation from Gaussian random variables.
- The writing is well-structured and precise.

# Weaknesses
The only minor weakness is that the writing is highly terse, and some parts end up appearing unmotivated or not quite justified. Some examples:
- In the proof of Theorem 3.4. It isn't obvious why the quantity in (27) is being bounded until much later, when a result from Shao and Zhang is invoked.
- The use of the Lindeberg-Feller CLT to prove Theorem 3.3 could be explained more thoroughly. It seems that it would follow from Assumption 2.2 and the construction of L, but more  details would be nice. (Namely, justifying the Lindeberg condition.)

A bit of restructuring to help the reader (and this reviewer!) understand the ideas behind of the proofs would be helpful.

---

> ### Author Rebuttal · Authors · 2025-07-31
>
> Thank you very much for your thoughtful evaluation and for recognizing the contributions of our work! Please see our point-to-point responses to your concerns below.
>
> $\textbf{Responses to Weaknesses}$
>
> $\textbf{1:}$ Thank you for your helpful and constructive suggestion! We totally agree that it is better to first invoke the result from Shao and Zhang, and to discuss some high-level ideas of each step. We will carefully rearrange the proof and the remark in our updated version.
>
> For instance, we would like to explain some motivations behind the functional dependence measure (Theorem 3.1) and the proofs of the main theorems.
>
> In the context of SGD, the functional dependence measure quantifies the contribution of the $t$-th training sample $X_t$ to the $n$-th SGD iterate $\theta_n$. In other words, it measures the degree of dependence of outputs $\theta_n$ on inputs $X_t$. Theorem 3.1 indicates that this degree of dependence decays exponentially fast for the constant learning rate SGD.
>
> The derivation of the Berry-Esseen inequality (Theorem 3.4) for SGD relies on the idea of linear approximation and coupling. Theorem 3.1 plays a vital role in the calculation of the dependence measure of the error terms from linear approximation, which constitute the final bound.
>
> The proof of Theorem 4.2 is inspired by (Nagaev 1979), who establishes a sharp concentration inequality of independent sums relying on the multiplicativity of moment generating functions (MGF). However, the non-linearity and dependence structure of SGD becomes a roadblock, where the MGF can not be explicitly computed. To address this issue, we leverage an autoregressive coupling idea, i.e., approximate the SGD iterates with a gradient descent iteration with i.i.d. noises (Lemma 4.1). The MGF of the coupling sequence is tractable, based on which we apply a Chernoff-type bound to obtain the result.
>
> $\textbf{2:}$ The multivariate Lindeberg-Feller CLT states the following:  Suppose
>
> $y_{nk} \in \mathbb{R}^d$
> are independent with means
>
> $\mathbb{E}(y_{nk})=0$ and variance matrices
>
> $$
> V_{nk} = \mathbb{E} ( y_{nk}y _{nk}^T ).
> $$
>
> Set
> $$U_n =  \sum_{i=1}^n V_{nk}, \nu_n^2 = \lambda\_{\min}(U_n).
> $$
> If $\nu_n^2 > 0$ and for all $\varepsilon > 0$
> $$
> \lim_{n \to \infty} \frac{1}{\nu_n^2} \sum_{k=1}^n \mathbb{E} \left( |y_{nk}  |^2
> \cdot \mathbf{1}\left( |y_{nk}  |^2 \geq \varepsilon \nu_n^2 \right) \right) = 0
> $$
> then as $n \to \infty$
> $$
> U_n^{-1/2}  \left( \sum_{k=1}^n y_{nk}  \right) \Rightarrow \mathcal{N}(0, I_d).
> $$
> We apply the theorem on the standardized sum
> $\Gamma\_n(\gamma)^{-1/2}\sqrt{\gamma} \sum_{k=1}^n (I_d-\gamma A)^{n-k}\ Z_k$.
> Denote
>
> $$B\_{nk}(\gamma)=\Gamma_n(\gamma)^{-1/2}\sqrt{\gamma} \sum_{k=1}^n ({\bf I}_d-\gamma A)^{n-k}.$$
>
>
> Then $y_{nk}=B_{nk}(\gamma) Z_k$, $U_n = I_d$ and $\nu_n^2=1$. Let $M_n(\gamma)$ be the maximum norm of the matrix prefactor of $y_{nk}$, i.e.,
>
> $$ M_n(\gamma)= \max\_{1\leq k \leq n}|B_{nk}(\gamma) |=\max\_{1\leq k \leq n} | \Gamma_n(\gamma)^{-1} \gamma ({\bf I}_d-\gamma A)^{2n-2k}|. $$
>
> At the end of page 16, we show that $ | \Gamma_n(\gamma)^{-1}|$ is bounded by some constant. As a result, $M_n(\gamma) \lesssim \gamma \rightarrow0$. This is sufficient for the Lindeberg condition and CLT. To this end, notice that
> $$
> \sum_{k=1}^n \mathbb{E} ( |y_{nk}  |^2
> \cdot \mathbf{1}  (|y_{nk}  |^2 \ge \varepsilon ))  \leq  \sum_{k=1}^n |B_{nk}(\gamma)|^2 \mathbb{E} ( |Z_k  |^2
> \cdot \mathbf{1} ( M_n(\gamma) |Z_k |^2 \ge \varepsilon )).
> $$
>
> We have
>
> $\mathbb{E} \left( |Z_k  |^2
> \cdot \mathbf{1}\left( M_n(\gamma) |Z_k |^2 \geq \varepsilon  \right) \right) \rightarrow 0$ due to $\mathbf{1}\left( M_n(\gamma) |Z_k |^2 \geq \varepsilon  \right) \rightarrow 0$
>
> and dominated convergence theorem. By elementary calculation, $\sum_{k=1}^n |B_{nk}(\gamma)|^2$ is bounded. As a result, the Lindeberg condition is justified.
>
> $\textbf{Responses to Questions}$
>
> $\textbf{Q1:}$ Yes, constants can be quantified. In our Theorem 2.3, Theorem 3.1 and Lemma 4.1, we specified all the constants. Lemma 3.2, Theorem 3.4 and Theorem 4.2 are based on those preliminary results, and the constants can be computed in a similar way. Since this paper mainly focuses on the convergence rate w.r.t. learning rate $\gamma$ and sample size $n$, and the expression of some constants could be tedious, we did not specify some of them in our theorems. We will provide explicit forms in our updated version of the appendix.
>      For example, in Lemma 3.2, we can take the constant as
>      $$ C_0 = \max\{\sqrt{\frac{3}{\alpha_0\lambda}}M_2\gamma+\frac{6L_2M_2^2}{\alpha_0\lambda} , \ \sqrt{\frac{2}{\lambda}}L_2\rho, \ 2L_2\rho_2^2 \}.$$
>      Here $\rho_2$, $M_2$ and $L_2$ are constants from Assumption 2.1 and 2.2.
>
> $\textbf{Q2:}$ Yes, it is possible. For example, [R1] proposes a piecewise Lyapunov function method to analyze SGD for non-strongly-convex problem. With stronger assumption on the tail behavior, our Theorem 2.3 can be extended to the non-strongly-convex setting following their idea. Other results are also extendable and will be an interesting future direction.
>
> $\textbf{Q3:}$ The constant in our Nagaev inequality can be rigorously justified by lengthy elementary calculations, as discussed in the reply to your second question. Our Nagaev-type bound is tight in the sense that the polynomial term $\gamma^{p-1}$ is much sharper than the Markov-type bound or other results in the literature. In this paper, we do not focus on optimizing these constants. However, we agree that it is an interesting and challenging future work.
>
>
> $\textbf{References}$
>
> [1] Zhang, Y., D. Huo, Y. Chen, and Q. Xie (2025). “A Piecewise Lyapunov Analysis of Sub-quadratic SGD: Applications to Robust and Quantile Regression”. In: ACM SIGMETRICS Performance Evaluation Review
> 53.1, pp. 85–87

---

> > ### Comment · Reviewer_AvJc · 2025-08-05
> >
> > This is a very nice paper, and the responses addressed my questions well. My score is already very high, and  I will keep it as is.

---

> > > ### Author Response · Authors · 2025-08-09
> > >
> > > We are delighted that all your questions have been addressed! Thank you again for your strong endorsement.

---

### Note · Authors · 2025-08-16

We would like to express our heartfelt gratitude to the reviewers, the AC, and the SAC for their dedicated time, insightful expertise, and thoughtful engagement with our work. We are grateful for the reviewers' unanimous support. They have reached a broad consensus that our paper is solid, novel, well-structured, and significant in both its theoretical contributions and practical implications. In particular, the reviewers recognized the following key contributions:

$\textbf{1.}$ A comprehensive study that establishes foundational asymptotic theory and sharp finite-sample guarantees of the constant learning-rate SGD algorithm. This paper presents rigorous theoretical analysis and elegant results on the $p$-th moment convergence, quenched central limit theorem, Berry-Esseen bounds for Gaussian approximation, and large-deviation inequalities.

$\textbf{2.}$ Introducing novel technical tools that could inspire further analysis in stochastic optimization, such as the functional dependence measure and the nonlinear autoregressive approximation.

$\textbf{3.}$ The precise quantitative properties that advance the understanding of the practical behavior of SGD and open up numerous downstream uncertainty quantification tasks.

We also appreciate the reviewers' constructive and helpful feedback that inspired meaningful improvements. We have addressed all concerns raised by the reviewers in our rebuttal discussion by providing further illustrations, explaining proof details, and outlining additional experimental results. In the revised manuscript, we will fully incorporate all the suggestions from the reviewers, including: (1) restructuring the presentation to highlight motivations and clarify proof strategies; (2) discussing potential relaxation of assumptions; (3) specifying the constants in the results and their dependence on other parameters such as the dimension $d$; and (4) demonstrating the sharpness of the Berry-Esseen bound via Edgeworth expansion.

We sincerely hope that our paper will be evaluated favorably for its originality, substantial contributions, and careful refinement based on the reviews. Thank you again for your valuable time and effort.

---

### Decision · Program_Chairs · 2025-09-17

**Decision:**

Accept (poster)

**Comment:**

This work develops a comprehensive theory for constant learning-rate SGD, establishing a quenched CLT with mild conditions, sharp Berry–Esseen bounds, and non-asymptotic large-deviation guarantees, thereby enabling rigorous uncertainty quantification.
Strength: The paper establishes an array of results for constant step-size SGD which are of interest to the broader machine learning community.

Weakness: The assumption of strong convexity is limiting. An important references [1], which establishes CLT for constant step-size SGD in the nonconvex setup, is completely missed in this work.

[1] - Yu, Lu, et al. "An analysis of constant step size SGD in the non-convex regime: Asymptotic normality and bias." Advances in Neural Information Processing Systems 34 (2021): 4234-4248.

The paper is unanimously recommended for acceptance by all reviewers, a decision which the AC also agrees with.